Applied and Environmental Science

# Continental-Scale Microbiome Study Reveals Different Environmental Characteristics Determining Microbial Richness, Composition, and Quantity in Hotel Rooms

Xi Fu,[a,c] Yanling Li,[a,b] Qianqian Yuan,[a,b] Gui-hong Cai,[d] Yiqun Deng,[a,b] Xin Zhang,[e] Dan Norbäck,[d] Yu Sun[a,b]

[a]Guangdong Provincial Key Laboratory of Protein Function and Regulation in Agricultural Organisms, College of Life Sciences, South China Agricultural University, Guangzhou, Guangdong, People's Republic of China

[b]Key Laboratory of Zoonosis of Ministry of Agriculture and Rural Affairs, South China Agricultural University, Guangzhou, Guangdong, People's Republic of China

[c]School of Public Health, Sun Yat-sen University, Guangzhou, People's Republic of China

[d]Occupational and Environmental Medicine, Department of Medical Science, University Hospital, Uppsala University, Uppsala, Sweden

[e]Institute of Environmental Science, Shanxi University, Taiyuan, People's Republic of China

Yanling Li and Qianqian Yuan contributed equally to this article.

**ABSTRACT** Culture-independent microbiome surveys have been conducted in homes, hospitals, schools, kindergartens and vehicles for public transport, revealing diverse microbial distributions in built environments. However, microbiome composition and the associated environmental characteristics have not been characterized in hotel environments. We presented here the first continental-scale microbiome study of hotel rooms ($n = 68$) spanning Asia and Europe. Bacterial and fungal communities were described by amplicon sequencing of the 16S rRNA gene and internal transcribed spacer (ITS) region and quantitative PCR. Similar numbers of bacterial (4,344) and fungal (4,555) operational taxonomic units were identified in the same sequencing depth, but most fungal taxa showed a restricted distribution compared to bacterial taxa. Aerobic, ubiquitous bacteria dominated the hotel microbiome with compositional similarity to previous samples from building and human nasopharynx environments. The abundance of *Aspergillus* was negatively correlated with latitude and accounted for ~80% of the total fungal load in seven low-latitude hotels. We calculated the association between hotel microbiome and 16 indoor and outdoor environmental characteristics. Fungal composition and absolute quantity showed concordant associations with the same environmental characteristics, including latitude, quality of the interior, proximity to the sea, and visible mold, while fungal richness was negatively associated with heavy traffic (95% confidence interval [CI] = −127.05 to −0.25) and wall-to-wall carpet (95% CI = −47.60 to −3.82). Bacterial compositional variation was associated with latitude, quality of the interior, and floor type, while bacterial richness was negatively associated with recent redecoration (95% CI −179.00 to −44.55) and mechanical ventilation (95% CI = −136.71 to −5.12).

**IMPORTANCE** This is the first microbiome study to characterize the microbiome data and associated environmental characteristics in hotel environments. In this study, we found concordant variation between fungal compositional variation and absolute quantity and discordant variation between community variation/quantity and richness. Our study can be used to promote hotel hygiene standards and provide resource information for future microbiome and exposure studies associated with health effects in hotel rooms.

**KEYWORDS** *Aspergillus*, environmental microbiology, hotel, indoor microbiome, microbial ecology

Address correspondence to Dan Norbäck, dan.norback@medsci.uu.se, or Yu Sun, sunyu@scau.edu.cn.

Recent advances in culture-free high-throughput sequencing techniques and bioinformatics analyses have greatly facilitated microbiome research in many fields, including human gut, skin, and respiratory tract and disease, and environmental microbiomes, such as earth's microbiome project (1–6). Although the total number of studies has increased dramatically in the past few years, they are unevenly distributed among areas. For example, more than 60% of the microbiome studies are restricted to the human gut and skin and laboratory-based model organisms, and only ~2% of the studies were performed in the built environment (7). The National Human Activity Pattern Survey from the United States reported that people spend an average of 87% of their time in buildings and another 6% of their time using transportation (8). Furthermore, indoor microbial exposures have been reported to relate to occupant health (9). Several risk and protective species have been identified to be associated with human diseases such as asthma, allergy, and respiratory symptoms (10–12), and exposure to high bacterial and fungal diversity has been reported to have protective effects on childhood asthma (13, 14). A long-term goal of indoor microbiome research is to identify a "healthy building microbiome" and promote human well-being, and the necessary first step is to characterize the microbiome composition in different indoor environments.

Indoor microbiome research is a complex multidisciplinary field that requires knowledge of microbiology, ecology, environmental science, building science, and epidemiology, as well as new techniques, such as next-generation sequencing (NGS) (15). To date, indoor microbiome studies have mainly focused on home environments (16–20), with several studies from other environments, such as hospitals (21, 22), schools (23), university dormitories (24), kindergartens (23), and vehicles for public transport (25, 26). In principle, the framework of the indoor microbiome, especially in the home environment, has been generally established. Indoor microbes originate from multiple sources, including outdoor air; soil; plants; human skin, gut, and mouth; pets; and plumbing systems (27). It is generally accepted that the outdoor environment and indoor occupants and animals are two primary sources of indoor bacteria (17, 28), but the relative contributions of the two sources vary with building type and location. For example, the percentage of indoor bacteria from human sources varied from 4% in a conference room (29) to >30% in a university housing complex (30). In a noncontaminated or moldy environment, indoor fungi are mainly sourced from outdoor air and thus structured by climate and geographical patterns (17, 20). Other indoor factors, such as mechanical ventilation, type of carpet, and cleaning procedure and frequency, also shape indoor fungal communities (10). Identifying the environmental factors associated with microbial composition promotes further understanding of indoor microbiome variations.

Hotels are common public environments for guests and hotel staff. There are at least 22 million hotel rooms in the world (31), and billions of guests and travelers stay in hotels each year. Thus, there is a public health concern regarding hotels' hygiene standards and practice. Unlike household residences, each hotel room is shared by many guests, and many environmental characteristics affecting the microbiome variation are controlled in the hotel environment. For example, many hotels use standard cleaning procedures and ventilation systems for air exchange (32), and no pets are allowed in hotel rooms, so a major source of the indoor microbiome is controlled. Therefore, hotel rooms are an appropriate place to conduct a global indoor microbiome comparison. A few studies have quantified hotel bacterial and fungal taxa by counting colony-forming microbes on medium (33, 34), but due to technical limitations, this approach can identify only <1% of total microbial species (35). Our previous study used quantitative PCR (qPCR) to monitor fungal quantity in hotels and identified several associated environmental characteristics (36). A few hotel epidemiological studies have focused on single infectious microbial exposure or outbreaks in hotel rooms, such as *Legionella* and norovirus infections (37, 38). Overall, since no microbiome survey has been conducted in the hotel environment, the overall assemblage and diversity of hotel microbes and their environmental drivers are still unknown.

Microbiome studies must quantify and disentangle thousands of phylogenetically related or distinct species, and $\alpha$- and $\beta$-diversity values represent popular statistics to describe microbial composition and distribution. The $\alpha$ diversity, including the number of observed species, the Chao1 index, and the Shannon index, quantifies the community richness within an individual sample (39–41). The $\beta$ diversity, including quantitative metrics such as the Bray-Curtis or UniFrac distance, evaluates the composition variations between samples (42, 43). Together with quantitative approaches, such as qPCR, the absolute number of microbial cells per taxon can be identified (44), which can be used to identify the microorganisms associated with health outcomes and environmental characteristics (14, 18, 45).

In this study, we sequenced microbial amplicon regions, including the bacterial 16S rRNA gene and fungal internal transcribed spacer (ITS), to characterize the microbiome composition of hotel dust in a large geographic area covering 19 European and Asian countries. In total, 16 environmental factors were analyzed together with microbiome data in multiple linear regression and permutation models to identify factors associated with microbial richness, compositional variation, and quantity. We further discussed the results and implications from microbial, ecological, and indoor health perspectives.

## RESULTS

**Microbial diversity and composition in hotel dust.** In this study, dust swab samples were collected on the top of doorframe from 68 hotels in 19 European and Asian countries (Fig. 1). The sampling location was rarely cleaned by hotel staff and thus reflected the airborne microbiome composition accumulated over at least several months. Two dust samples were collected in each room. One was used for quantitative analysis of fungi DNA, the results of which were published in a previous study (36), and the other sample was used for 16S rRNA gene and ITS region sequencing in the present study. Sixteen environmental characteristics were recorded for the microbiome association analysis (see Table S1 in the supplemental material).

Based on gel electrophoresis of the negative control, no DNA contamination in the reagents or amplification processes was observed (Fig. S1). Several quality-control and filtration steps were conducted for the sequence data (see Materials and Methods). The rarefaction analysis indicated that the sequencing depth was sufficient to cover the majority of the microbial diversity (Fig. S2). The bacterial and fungal sequencing data were both rarefied to the depth of 27,000 reads, and all following analyses were conducted at this depth. A total of 4,344 bacterial and 4,555 fungal operational taxonomic units (OTUs) were obtained, and each dust sample harbored 988 bacterial and 370 fungal OTUs on average. Bacterial and fungal samples had distinct OTU distribution patterns (Fig. 2A and B). The bacterial OTUs were more widespread on the continental scale; more than half of the OTUs were found in 10 or more samples, and 43 OTUs were found in all samples. The most widely spread bacterial OTUs were also mainly the high abundant bacterial OTUs, such as *Pelomonas* and *Ralstonia*. Most fungal OTUs were present in very few samples; approximately half of the OTUs were found in one or two samples. The widely spread fungal OTUs were also the highly abundant fungal OTUs. *Candida albicans* was found in all samples, and the mean abundance of the species was 4.1%. *Aspergillus* was found in 66 samples, and the mean abundance of the genus was 26.5%.

Taxonomic information is presented at the phylum and genus levels based on search results from the Silva database (Fig. 2C and Fig. S3). For bacteria, the dominant phylum was *Proteobacteria* (mean, 71.8%), followed by *Firmicutes* (10.7%) and *Actinobacteria* (6.5%) (Table S2). The top genera were the environmental bacteria *Ralstonia* and *Pelomonas* (>10%), followed by *Cupriavidus*, *Ochrobactrum*, *Acinetobacter*, *Brevundimonas*, *Anoxybacillus*, *Sphingomonas*, *Pseudomonas*, *Geobacillus*, and *Corynebacterium* (Table S2). We searched the bacterial compositional of our samples against a curated public database Microbiome Search Engine, which contains the compositional information of more than 230,000 bacterial microbiome samples across a variety of environments (46). The engine calculates a score for a query sample, and a high novelty

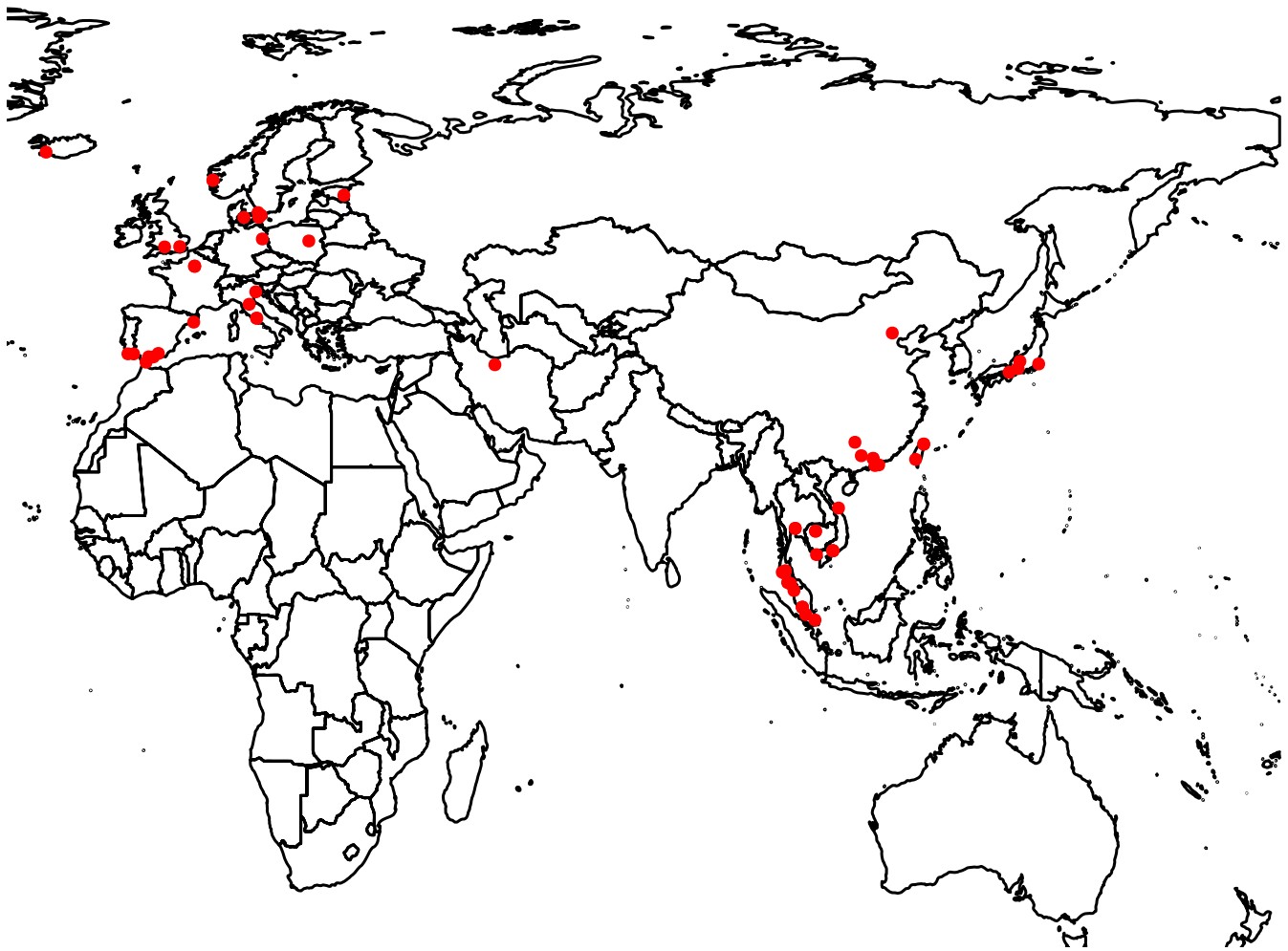

**FIG 1** Dust sampling locations in Asia and Europe.

score (>0.12) represents a low similarity to previous samples in the database. The similarity scores were >0.12 for 70% of the hotel samples, indicating high compositional novelty in our hotel bacterial data set (Table S3). The most similar bacterial microbiome samples were mainly from building and human nasopharynx and skin environments.

The dominant fungal phyla were Ascomycota (78.3%) and Basidiomycota (15.4%; see Fig. S3 and Table S2 in the supplemental material). The most abundant genus was *Aspergillus* (>25%), followed by *Mycosphaerella*, *Candida*, *Aureobasidium*, and several unidentified Ascomycota (Fig. 2D and Table S2). The top fungal genus *Aspergillus* had an uneven distribution among samples, with an increasing trend from high to low latitude (Fig. 2D). The relative abundance of *Aspergillus* was over 80% in eight hotels, and seven of which were from low latitudes, including four in Malaysia, two in Thailand, and one in Vietnam. For the high-latitude hotels, the abundance of *Aspergillus* was lower than 15%, except in Venice (Fig. 2D).

**Associations between environmental characteristics and hotel microbiome.** Associations between environmental characteristics and microbiome, including bacterial and fungal richness and compositional variation, were quantitatively analyzed (Table 1). The absolute quantification of fungal DNA was also included in the association analyses, which was assessed by quantitative PCR by two sets of primers in our previous studies (36). The two primers targeted different fungal amplicon regions (ITS1 and 28S rRNA gene) and captured a wide range of indoor fungal species. We first

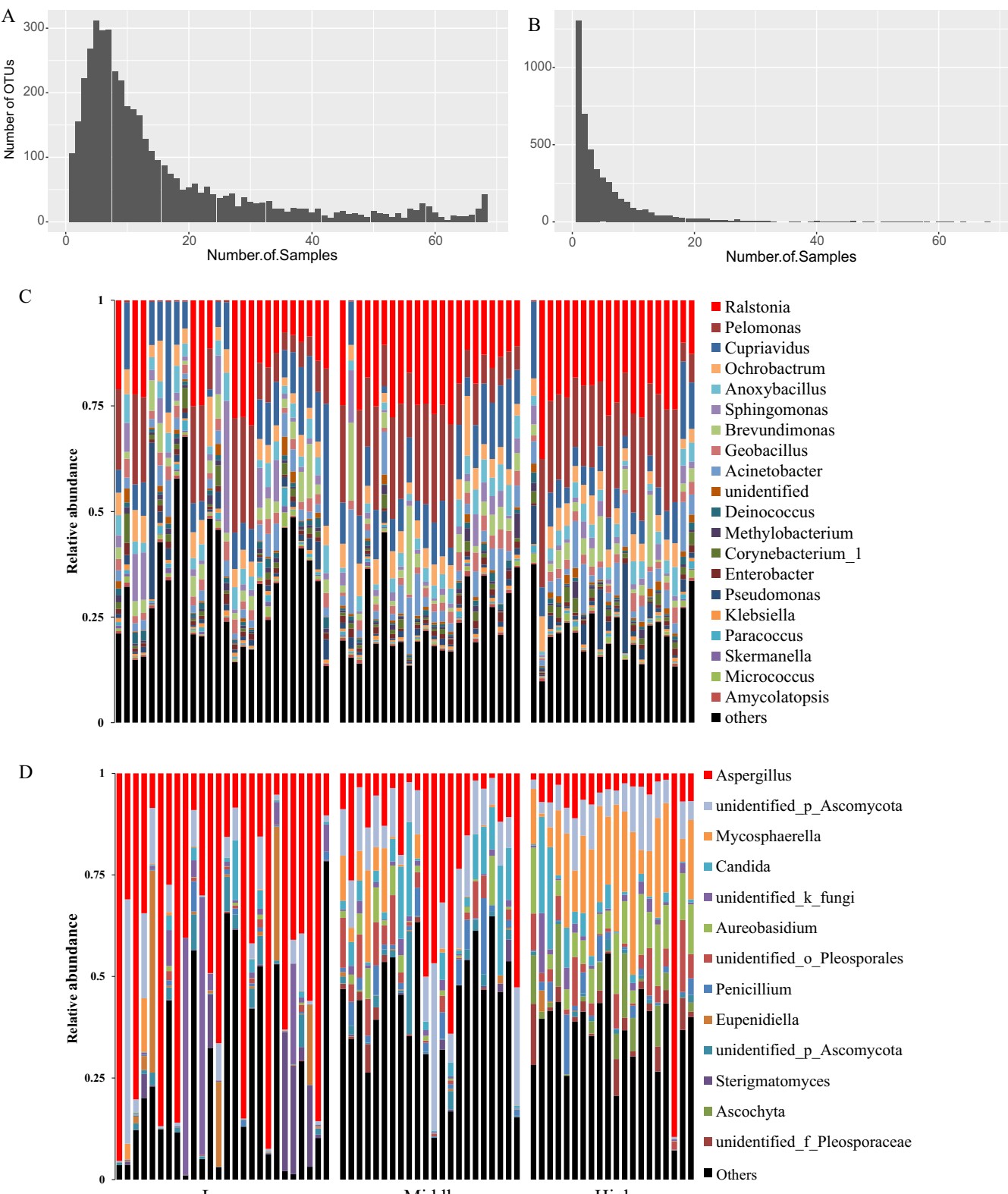

**FIG 2** Distribution and relative abundance of bacterial and fungal taxa in hotels. (A and B) Frequency spectra for the number of bacterial (A) and fungal (B) OTUs in samples. (C and D) Relative abundances of top bacteria (C) and fungi (D) at the genus level for all samples. Hotels in low, middle, and high latitudes were placed in the left, middle, and right parts along the x axis.

**TABLE 1** Multivariate analysis between outdoor/indoor characteristics and microbial richness and compositional variation and fungal quantity[a]

| Characteristic | Bacteria | | Fungi | | | |
| --- | --- | --- | --- | --- | --- | --- |
| | Observed OTU, beta (95% CI) | Community variation ($R^2$) | Observed OTU, beta (95% CI) | Community variation ($R^2$) | Fungal DNA 1, beta (95% CI) | Fungal DNA 2, beta (95% CI) |
| Climate/outdoor characteristics | | | | | | |
| Latitude | | **0.10\*\*\*** | | **0.14\*\*\*** | | **–0.40 (–0.62, –0.18)\*\*** |
| Proximity to the sea | | | | **0.03\*** | **0.82 (0.37, 1.27)\*\*** | **0.89 (0.39, 1.38)\*\*** |
| Proximity to roads with heavy traffic | | | **–63.65 (–127.05, –0.25)\*** | | | |
| Hotel/indoor characteristics | | | | | | |
| No. of yrs since redecoration[b] | **–111.77 (–179.00, –44.55)\*\*** | | –56.67 (–115.05, 1.72) | | | |
| Size of the hotel | | | | | 0.25 (–0.04, 0.54) | |
| Floor level[c] | | | | | **–0.21 (–0.39, –0.04)\*** | |
| Quality of the interior[d] | | **0.04\*** | | 0.02 | **0.39 (0.19, 0.59)\*\*\*** | **0.26 (0.04, 0.48)\*\*** |
| Dampness or mold | | | | 0.02 | **0.47 (0.09, 0.85)\*** | **0.77 (0.31, 1.23)\*\*** |
| Mechanical ventilation | **–70.92 (–136.71, –5.12)\*** | | | | | |
| Floor type[e] | | **0.06\*\*** | **–25.71 (–47.60, –3.82)\*** | 0.02 | | |
| Sum $R^2$ | 0.19 | 0.20 | 0.20 | 0.22 | 0.56 | 0.64 |

[a]The microbial richness and fungal quantity were calculated by a forward stepwise linear multiple regression, and the microbial community was calculated by a forward stepwise Adonis multivariate analysis with 10,000 permutations. Only environmental characteristics with $P < 0.1$ were kept in the final multivariate model. Associations with $P$ values of $<0.05$ are indicated by boldfacing and asterisks (\*\*\*, $P < 0.001$; \*\*, $P < 0.01$; \*, $P < 0.05$), and associations with $0.05 < P < 0.1$ are presented in regular typeface. Observed OTU values ($\alpha$ diversity) and fungal DNA 1 and 2 values ($\alpha$ diversity) are expressed as beta (95% CI); community variation values ($\beta$ diversity) are expressed as $R^2$. The sums of $R^2$ values of the final multivariate model are also presented in the last row. Fungal DNA 1 was estimated by qPCR of the ITS1 region which captured at least 530 fungal species; fungal DNA 2 was estimated by qPCR of the 28S rRNA region which captured at least 140 fungal species. A detailed list of the fungal species captured was provided in previous publications (36, 80). A/C, air conditioning.
[b]Number of years since redecoration: 0, $>5$ years; 1, $<5$ years.
[c]Floor level: 1, ground; 2, top; 3, 2 to 4 floors; 4, 5 to 22 floors.
[d]Quality of the interior: 1, high; 2, normal; 3, old.
[e]Floor type: 1, stone; 2, wood; 3, plastic; 4, wall-to-wall carpet.

conducted the bivariate analyses between environmental characteristics and microbiome data to screen for potential associated environmental characteristics (Table S4). The environmental characteristic with $P$ value of $<0.2$ in the bivariate analysis were included in the final multivariate analysis model by the forward stepwise approach (Table 1). The Kruskal-Wallis test conducted the associations between environmental characteristics and bacterial/fungal richness (based on the number of observed OTUs) and fungal DNA. The multidimensional microbial composition data cannot be analyzed by simple statistics such as the Kruskal-Wallis test and thus were analyzed by Adonis (47).

For bacteria, latitude was the strongest factor associated with compositional variation ($P < 0.001$, $R^2 = 0.10$, Adonis). Floor type and quality of the interior were weakly associated with bacterial compositional variation ($P < 0.05$, $R^2 = 0.06$ and 0.04, respectively). Recent redecoration and the presence of mechanical ventilation were negatively associated with bacterial richness in the hotel rooms (linear regression coefficient beta = –111.77, 95% confidence interval [CI] = –170.00 to –44.55; beta = –70.92, 95% CI = –136.71 to –5.12). Thus, bacterial richness and compositional variation were associated with different environmental factors.

For fungal, latitude was also the strongest predictor of compositional variation ($P < 0.001$, $R^2 = 0.14$, Adonis) and was significantly associated with fungal quantity (beta = –0.40, 95% CI = –0.62 to –0.18, linear regression). Proximity to the sea, low quality of

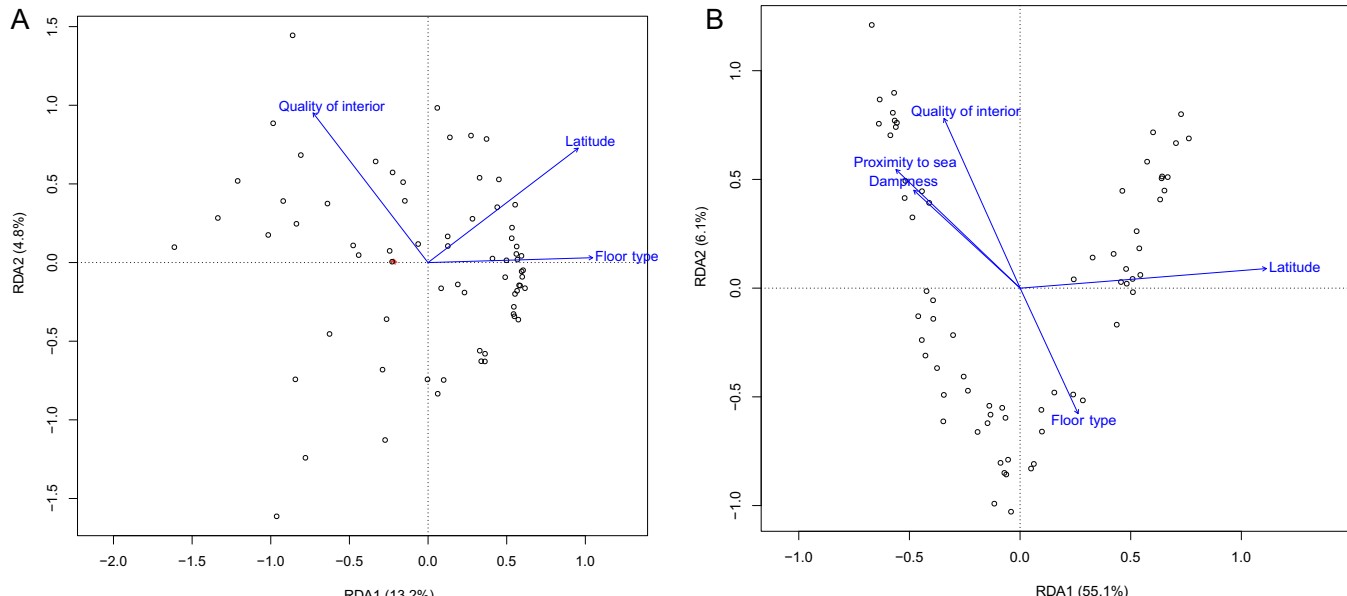

**FIG 3** Redundancy analysis (RDA) of bacterial (A) and fungal (B) composition. Environmental characteristics associated with compositional variation identified by Adonis were projected in the plot.

the interior, and visible mold were weakly associated with fungal community variation ($P = 0.03$, 0.07 and 0.07 and $R^2 = 0.03$, 0.02, and 0.02, respectively) and positively associated with fungal DNA quantity (beta = 0.82, 95% CI = 0.37 to 1.27; beta = 0.39, 95% CI = 0.19 to 0.59; and beta = 0.47, 95% CI = 0.09, 0.85 [respectively]). However, these characteristics did not affect fungal richness. Two factors were weakly associated with fungal richness. The floor type was negatively associated with fungal richness (beta = −25.71, 95% CI = −47.60 to −3.82). Proximity to roads with heavy traffic was negatively associated with fungal richness (beta = −63.65, 95% CI = −127.05 to −0.25). Thus, the factors associated with fungal community variation were also associated with absolute quantity, while fungal richness was affected by other environmental characteristics.

The effects of environmental characteristics on the microbial community variation were further illustrated by the redundancy analysis (RDA; Fig. 3). The environmental characteristics associated with bacterial and fungal community variation ($P < 0.1$, Adonis) were projected on the plot. Latitude was a variable explaining bacterial variation along RDA axes 1 and 2. Floor type explained bacterial variation along RDA axis 1. For the fungal community, latitude was an important characteristic explaining variation along RDA axis 1, which accounted for 55.1% of eigenvalues. Other environmental factors, such as floor type and quality of the interior, mainly explained fungal variation along RDA axis 2.

**Environmental characteristics and difference in microbial abundance.** We further characterized and visualized bacterial and fungal genera with different abundance between environmental conditions (Fig. 4). For bacteria, *Ralstonia* and *Pelomonas* were significantly more abundant at high latitudes (Kruskal-Wallis test with the Benjamini-Hochberg correction, false-discovery rate [$q$] < 0.05), while *Cupriavidus* and *Saccharopolyspora* were more abundant at low latitudes ($q < 0.05$). For fungi, *Aspergillus*, *Eupenidiella*, *Sterigmatomycetes*, *Schizophyllum*, and *Exobasidium* were more abundant at low latitudes ($q < 0.05$), while *Mycosphaerella*, *Aureobasidium*, *Penicillium*, *Malassezia*, *Cryptococcus*, *Simplicillium*, *Botrytis*, *Stemphylium*, *Ascochyta*, and two unidentified genera were more abundant at middle or high latitudes ($q < 0.05$).

*Saccharopolyspora* was the only bacterial genus associated with the quality of the room interior (textiles, walls, and furniture), and the genus was more abundant in worn and old rooms (Fig. S4). The relative abundance of *Aspergillus* was 54.9% in rooms with

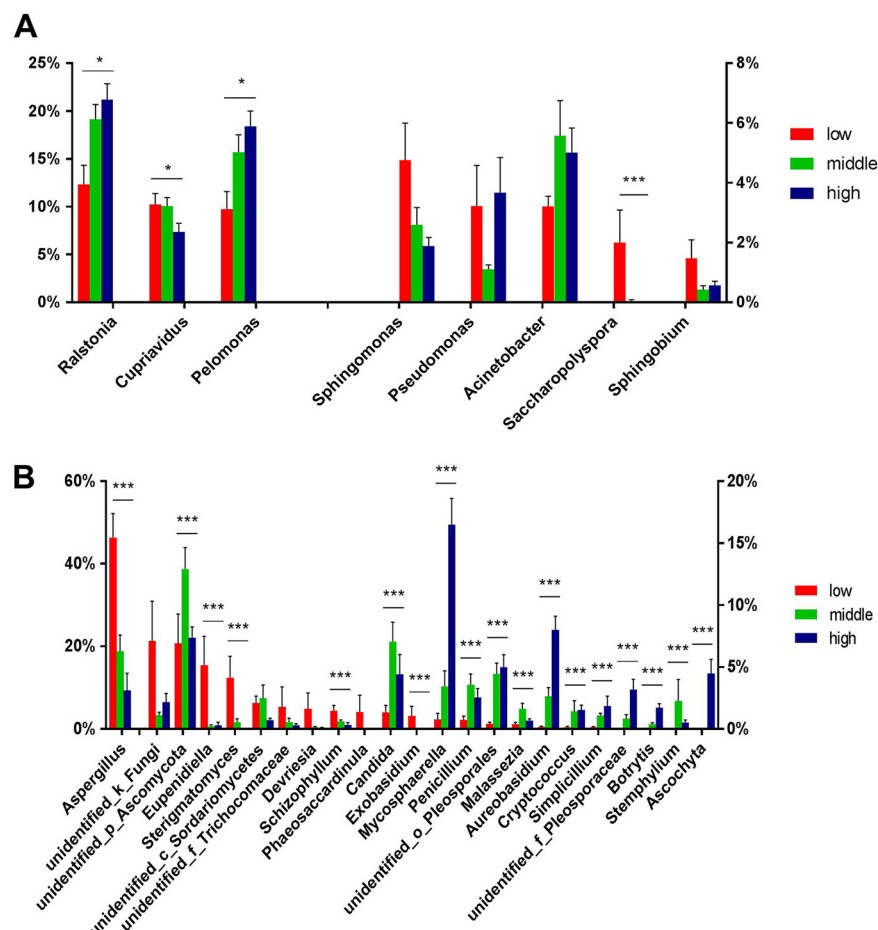

**FIG 4** Relative abundance of bacterial (A) and fungal (B) genera at different latitudes. Only genera with relative abundance differences of >1% among latitudes are plotted. Error bars represent the standard errors, and a Kruskal-Wallis test with a Benjamini-Hochberg correction was conducted to calculate the *P* values (***, *q* < 0.001; **, *q* < 0.01; *, *q* < 0.05).

visible mold or dampness but only 19.2% in rooms without mold (*q* < 0.05; Fig. S5), which is consistent with the fact that *Aspergillus* is a common mold in the indoor environment. *Eupenidiella* and *Exobasidium* were also more abundant in moldy hotel rooms (*q* < 0.05; Fig. S5).

Previous studies reported that the absolute quantification approach is more accurate than the relative quantification approach in identifying the associations between microbes and phenotypes (44). Thus, we conducted the same analysis between environmental characteristics and specific microbes based on the absolute quantification, which was calculated as multiplying the relative abundance of the fungi and qPCR data (Table S5). We confirmed that all significant differences detected by the relative abundance were also detected by the absolute abundance approach.

## DISCUSSION

Previous studies mainly characterized the microbial richness (*α* diversity) and compositional/community variation (*β* diversity) and quantity in the home environments. For example, a microbiome study in southern New England revealed that microbial richness was associated with the presence of pets, water leaks and suburban locations, and microbial compositional variation was associated with air conditioner usage and occupancy (18). The results suggest that microbial *α*- and *β*-diversity values are associated with different environmental characteristics in home environments, consistent with our results. There are also several continental-scale microbiome studies in the

home environments, revealing the importance of geographic distance and outdoor factors in structuring fungal composition (17, 48). A study of a university housing facility showed that season was an important factor shaping microbial α- and β-diversity values but was not associated with indoor fungal quantity, and geographic distance shaped fungal community variation (49). In this study, we reported the first microbiome survey in the hotel environment and found the concordant/discordant pattern for factors associated with microbial richness, compositional variation and quantity.

**High-abundance bacterial and fungal taxa in hotel rooms.** Most of the high-abundance bacteria in hotel rooms are ubiquitous taxa in outdoor environments and are widespread in a wide range of habitats, such as *Ralstonia*, which is commonly found in soils, rivers, and lakes, and *Pelomonas*, which is found in water and soil (50). Due to their ubiquitous nature, these bacteria are also frequently reported as sources of contamination for reagents and laboratories that impact the accuracy of sequence-based microbiome analyses (51). The negative control in the amplification process indicates no mass DNA or microbial contamination in this step. We also checked the sequencing projects in the same MiSeq run and found that most samples did not contain *Ralstonia* or *Pelomonas* species (data not shown), suggesting these taxa were not derived from laboratory contamination during the sequencing process.

A previous study reported that human-associated microbes, especially those on the skin, were an important source of the indoor microbiome, accounting for 4 to 40% of the total bacterial load (28). In the hotel dust samples, the abundance of human-associated microbes, such as *Acinetobacter* (4.5%), *Propionibacterium* (3.2%), *Corynebacterium* (1.4%), *Streptococcus* (0.25%), *Staphylococcus* (0.33%), *Bifidobacteria* (0.21), and *Kocuria* (0.41%), was lower than that of the environmental taxa. The total contribution of human-associated taxa in hotels was approximately 10 to 15%. The proportion is very similar to previous dust samples collected from the upper door trim (11%) and indoor air (11%) (17, 52). However, for dust collected from sampling sites with frequent human contact, such as a doorknob or bed, the proportion of human-associated taxa can be much higher (16, 28).

In hotel rooms, *Aspergillus* was the dominant fungus, accounting for on average one-fourth of the total fungal load. We also checked other sequencing projects in the same MiSeq run and did not find a high abundance of *Aspergillus*, indicating it is unlikely to be derived from laboratory contamination. The species was not evenly distributed among all samples; in 13 hotels, the abundance of *Aspergillus* was >50%, and in approximately half of the hotels, the abundance was <10%. The abundance of *Aspergillus* was significantly higher in low-latitude hotels (low-latitude mean, 46.3%; middle-latitude mean, 18.8%; high-latitude mean, 9.3%). The result is consistent with a previous global indoor fungal survey that *Aspergillus* was detected more often in tropical countries, such as Mexico and Indonesia, compared to high-latitude countries as Canada (48). The drastic variation in the abundance of *Aspergillus* in the indoor environment has also been reported. A previous study showed that the abundance of *Aspergillus* could vary dramatically from 0 to >95% in 1,200 homes across the United States (17). In addition to low latitude, proximity to the sea and visible mold were also associated with a high abundance of *Aspergillus*. These environmental characteristics are suggested to relate to high indoor air moisture and relative humidity (RH) (10). RH is a key factor in regulating fungal growth in the laboratory. Incubation of room dust at an 84 to 86% RH resulted in a 45-fold increase in *Aspergillus* and *Penicillium* (53). A chamber study demonstrated that *Aspergillus* and *Wallemia* growth occurred at >80% RH on carpets after 1 week of incubation (54). *Aspergillus* can produce fungal fragments, such as pieces of spores or hypha, microbial volatile organic compounds, or mycotoxins, and lead to various allergic or inflammatory symptoms in occupants, such as cough, wheezing, and headaches (10, 55, 56). Allergens of *Aspergillus fumigatus* have been extensively characterized by IUIS Allergen Nomenclature Committee. These allergens, including Asp f 36, Asp f 37, Asp f 1, and many others, can cause type I and type III hypersensitivity reactions in humans (57). The allergic effects of other *Aspergillus*

species, such as *Aspergillus flavus* and *Aspergillus niger*, were also reported (58). Thus, it may be necessary to routinely monitor the mold growth and *Aspergillus* quantity in hotel rooms, especially at low latitudes.

It is worth noting that some commonly reported airborne fungal genera (10, 48) are not present in high abundances in hotels, such as *Penicillium* (2.2%), *Cladosporium* (0.26%), *Acremonium* (0.44%), *Alternaria* (0.06%), *Fusarium* (0.08%), *Mucor* (<0.001%), *Stachybotrys* (0.12%), *Trichoderma* (0.32%), and *Trichophyton* (<0.001%). In contrast to *Aspergillus*, *Penicillium* was more abundant in the middle and high latitudes (3.6 and 2.5% versus 0.7%, $P < 0.001$). Previous studies identified *Penicillium* as one of the major airborne fungi in buildings that were mainly sampled in the middle- or high-latitude countries, such as Norway, France, and Poland (59–61). A global-scale sampling of settled dust revealed that *Penicillium* was present in low abundance in low-latitude indoor environments (48), which is also consistent with our observations in hotels.

**Factors associated with microbial richness in hotel rooms.** The latitudinal diversity gradient theory states that biodiversity declines with latitude, and the theory is supported by the majority of ecological studies (62, 63). However, deviation from this pattern has also been reported (64, 65). For example, an indoor study reported higher fungal diversity in temperate zones than in the tropics (48). Hillebrand conducted a meta-analysis of 600 studies and found that the strength of the diversity gradient increased significantly with organismal body mass, possibly due to energy use and dispersal limitation (62). Thus, microbial richness is less affected by latitude than the richness of larger vertebrates, which is consistent with the results we observed in the hotel data set.

A relationship between mechanical ventilation and fungal quantity indoors has been reported in many studies. Mechanical ventilation equipped with good filters can remove coarse airborne particles from outdoors, which was reported to reduce the indoor fungal concentration (10, 66). However, improperly maintained ventilation systems could also act as sources of contamination and increased fungal concentration (67). Mechanical ventilation is suggested to be associated with the variation of bacterial composition (68), but in this study, we found that it was negatively associated with bacterial richness but not associated with community variation. Thus, the role of mechanical ventilation is complex and may vary for different ventilation systems and building designs. We also found an association between recent redecoration and lower bacterial richness in the hotel rooms ($P < 0.01$, 95% CI = −179.00 to −44.55), which has not been reported in previous studies.

**Factors structuring microbial composition variation.** In this study, latitude was a strong factor determining both bacterial and fungal composition. The importance of latitude and geographic distance has been well documented in several microbiome studies (48, 49, 68). For example, a national microbiome survey revealed that both the bacterial and fungal composition was significantly affected by geographic range and latitude (20). Also, it has been suggested that the fungal community showed stronger geographic patterns compared to the bacterial community (49). We confirmed this finding by showing that most of the fungal OTUs had restricted distribution and presented in only one or two hotel rooms. It has been reported that the source of indoor fungi is mainly derived from the outdoor environment, whereas the source of indoor bacteria is more complex and affected by both the outdoor environment and indoor occupants, pets, and plants (69). Thus, the bacterial distribution in the indoor environment can be facilitated by human travel and movement.

The importance of global and outdoor environmental characteristics in shaping the indoor fungal composition is well supported, but the importance of indoor characteristics is still under debate. Some studies suggest that indoor characteristics are not important. A survey of indoor environments revealed that global factors, rather than building design and materials, determine the indoor fungal composition, and thus the indoor fungal assemblage represents a subsample of the outdoor fungal community (48). Other studies support this by showing that most fungi may not grow or proliferate

in the indoor environment; thus, the indoor environment mainly serves as a passive collector for the outdoor fungal biome (48, 49). However, some studies suggest that indoor characteristics are important in shaping indoor fungal composition. A recent survey of university residences in California found that fungal composition was clustered by indoor surface type, suggesting that some fungal species do grow or adhere to certain surface types (69). In this study, we found that indoor characteristics, such as the quality of the interior and floor surface type, were involved in shaping fungal composition in the indoor environment, supporting the latter hypothesis.

**Characteristics not associated with microbial diversity and variation.** Urban/ rural location was reported to be associated with microbial diversity or quantity (10). Farming environments have more diverse fungal resources than urban areas, which reduces early childhood asthma in rural areas (13). In this study, we did not detect an association between urban/rural locations and microbial variations. The association was significant in the bivariate analysis but not after adjusting for latitude and proximity to the sea in the multivariate model. This could be due to the medium collinearity between the hotel's proximity to the sea and urban/rural locations ($\rho = -0.54$, $P <$ 0.001, Pearson's correlation). Other hotel characteristics, such as the number of stars, building age, and size of the hotel, were not associated with microbial diversity and composition. Thus, factors associated with high-ranking hotels do not change the microbiome community in rooms.

**Study strength and limitations.** The strength of this study is that it is the first hotel microbiome study spanning large continental regions in Asia and Europe, providing useful resources for future indoor or hotel microbiome studies. The environmental metadata of the study were collected by a professional hygienist. Sampling was also performed in a standardized way, by a person with education on environmental sampling and inspections. Thus, the environmental characteristics were collected more consistently than the self-reported observations in home residence.

One limitation of this study is that samples were collected from one site in each hotel room. Previous studies showed that the sampling surface and indoor location affected microbial community composition (16, 70); thus, sampling dust at multiple sites can lead to a more comprehensive assessment of indoor microbiome. As the floor surfaces in hotels are frequently cleaned, it is not a good choice to investigate the hotel microbial exposure in floor dust. Active sampling, such as air vacuum pump and BioSampler, can sample airborne dust at inhalable heights, which might be a good sampling strategy to characterize short-term microbial exposure in hotel rooms for future studies. In this study, dust was collected by cotton swabs with a swabbing area of $1 \times 60$ cm for each sample; thus, the quantitative estimates were presented as the number of fungi per square meter. Since the hand pressure for swabbing may vary across the sampling sites, biases can be introduced in the sampling process. But we argue that the bias should be relatively small as the dust swabs were all collected by a single hygienist, and thus the sampling practice should be relatively consistent.

**Conclusions.** We presented here the first continental-scale hotel microbiome study spanning 19 countries and revealed the microbial composition and diversity. It is the first study to show that the environmental factors associated with the fungal community variation are also associated with the absolute quantity but not associated with fungal richness. *Aspergillus* was the most abundant fungus in hotel rooms and was negatively associated with latitude, whereas *Penicillium* was much less abundant, especially in low-latitude hotels. Most microbes in hotel rooms were ubiquitous species sourced from outdoor environments instead of from human sources. We uploaded all the data to the QIITA platform to facilitate research progress on the built environment. In the long term, these data can be integrated into a meta-analysis study to study human microbial exposure and promote human well-being in a general indoor environment.

## MATERIALS AND METHODS

**Sample and data collection.** Dust swab samples were collected by a professional hygienist, an academic person specialized in environmental sampling. Samples were collected from 68 hotels in 19 European and Asian countries from October 2007 to May 2009. Hotel rooms were arbitrarily chosen when checking in with no special request about the room. We collected dust samples in one room in each hotel by using dry cotton swabs to swab the upper half of the doorframe. The swab was designed for medical DNA sampling, and each swab was packaged in a DNA free and sterile plastic vessel (Copan Innovation, Brescia, Italy). Two samples were collected in each room with a swabbing area of 30 cm² (1 × 30 cm) on the left- and right-hand sides of the doorframe. Each swab was rotated slowly and moved back and forth three times over the surface. The swabs were stored in a −80°C freezer after sampling. One swab was used for amplicon sequencing in this study, and one swab was used for qPCR in a previous study (36).

Twenty-eight environmental characteristics were assessed and recorded at each hotel (Table S1). Eight environmental characteristics, such as a sign of flood and hot spring in the hotel, were presented in fewer than five hotels and thus removed from further analysis. Pearson's correlation was conducted to detect and reduce collinearity ($\rho > 0.7$), and four environmental characteristics were removed. For example, annual precipitation was highly correlated with latitude ($\rho = -0.82$); thus, latitude was kept for further analysis. A final set of sixteen environmental characteristics were kept for further analysis (Table S1), including latitude, surrounding traffic (heavy or light traffic), distance to an airport, proximity to the sea, location of the sampling site (rural, suburban, urban, or megacity), number of stars of the hotel, building age of the hotel, redecoration age of the hotel, size of the hotel, quality of the interior, floor level, visible dampness or mold in the room (yes/no), mechanical ventilation, floor surface type in the hotel room, air conditioner in the wall, and odor in the hotel room.

**Microbial DNA extraction and amplicon sequencing.** Total genomic DNA was extracted by an E.Z.N.A. Soil DNA kit D5625-01 (Omega Bio-Tek, Inc., Norcross, GA), which uses bead beating and spin filter technology to extract DNA. Total fungal DNA was extracted by Fast DNA SPIN extraction kits (MP Biomedicals, Santa Ana, CA). The quality and quantity of extracted DNA were evaluated by agarose gel electrophoresis, a NanoDrop ND-1000 spectrophotometer (Thermo Fisher Scientific, Waltham, MA), and a microplate reader (BioTek, FLx800), and all 68 samples passed the quality-control step and thus qualified for amplicon sequencing. The library was prepared by a TruSeq Nano DNA LT Library Prep kit from Illumina. The universal forward primer 338F (ACTCCTACGGGAGGCAGCA) and reverse primer 806R (GGACTACHVGGGTWTCTAAT) (71) were used for bacterial 16S rRNA gene V3-V4 region amplification, and the amplification region was 480 bp in length. The forward primer ITS5 (GGAAGTAAAAGTCGTAACAAGG) and reverse primer ITS2 (GCTGCGTTCTTCATCGATGC) (72) were used for fungal ITS1 region amplification, and the amplification region length was 250 bp. Sample-specific 7-bp barcode sequences were incorporated into primers for multiplex sequencing. Before sequencing, the library was evaluated by an Agilent Bioanalyzer and a Promega QuantiFluor with a Quant-iT dsDNA assay kit. Multiplex paired-end sequencing was performed according to the manufacturer's instructions. The sequencing was conducted in the Illumina MiSeq platform and a MiSeq reagent kit v3 (600 cycles) at Shanghai Personal Biotechnology Co., Ltd. (Shanghai, China).

**Bioinformatics and sequence analysis.** Raw sequences were extracted according to the barcode sequence and assigned to the respective samples. The raw sequences with a short length (<150 bp), low Phred score (<20) and ambiguous bases and mononucleotide repeats longer than 8 bp were removed (73). In total, 94.3% of bacterial raw reads and 97.9% of fungal raw reads passed the quality control filtering. Flash (v1.2.7) was used to assemble the paired-end reads with a minimum overlap between forward and reverse reads of >10 bp and no mismatches (74). Many of the following analyses were conducted with the Quantitative Insights Into Microbial Ecology (QIIME, v1.8.0) platform (75) and R packages. Chimeric sequences were removed by USEARCH (v5.2.236) (76). It has been shown that the erroneous reads from PCR in the amplicon preparation step and sequencing error lead to the overestimation of microbial diversity (77). Thus, we conducted a stringent quality-filtering step to extract high-quality data and set the OTU threshold (c value) to 0.01% in QIIME and other parameters following a previous suggestion (77). The remaining high-quality sequences were clustered into OTUs with 97% sequence identity by UCLUST (76). A representative sequence was picked for each out by pick_open_reference_otus.py in QIIME (v1.8.0) and blasted against the Silva database (release 115) (78) for bacteria and UNITE database (release 5) (79) for fungi to obtain taxonomic classification information. For the case of multiple best hits, the sequence was annotated with the taxonomy corresponding to the lowest common ancestor. The rounded rarefied analysis was conducted to standardize the sequencing depth to 27,000 reads per sample for both bacterial and fungal sequencing data. The richness index of the observed species was calculated based on the OTU table. Bivariate and multivariate linear regression and a Kruskal-Wallis test were performed by IBM SPSS Statistics (v21.0). In the multivariate linear regression model, an automatic forward stepwise approach was applied to include environmental characteristics with $P < 0.2$ in the bivariate analyses. Permutational bivariate and multivariate analysis of variance (Adonis) (47) was conducted by the vegan package in R with 10,000 permutations. The distance metrics were UniFrac distance metrics for bacteria and Bray-Curtis dissimilarity matrix for fungi (43). The multivariate Adonis analysis was calculated with a forward stepwise approach. The environmental characteristic with the lowest P value in the bivariate analysis was input first in the multivariate model, and the characteristic with the second lowest P value was input next, and so on (inclusion level $P = 0.2$). If the newly added characteristic did not improve the model ($P > 0.1$), the characteristic was removed from the multivariate model, and the next characteristic was tested. The world map was plotted by the "rworldmap" package, and RDA analysis was conducted by "vegan" package in R.

**Quantitative PCR of fungal DNA.** Our previous study used two sets of primers to quantify absolute fungal DNA in these hotel rooms (36). The first primer set targeted the fungal ITS1 region and captured a wide range of indoor fungi (>530 species), including 7 *Acremonium*, 61 *Alternaria*, 86 *Aspergillus*, 38 *Cladosporium*, 14 *Curvularia*, 27 *Eupenicillium*, 8 *Fusarium*, 17 *Neosartorya*. 15 *Paecillomyces*, 157 *Penicillium*, 9 *Rhinocladiella*, and several other species. The complete list of targeted species can be referred to (80). The second primer set targeted fungal 28S rRNA and captured mainly 37 *Aspergillus*, 62 *Penicillium*, 14 *Eupenicillium*, and several other species (in total >140 species). We named the two fungal quantifications fungal DNA 1 and fungal DNA 2 in this study.

**Data availability.** Sequencing data were deposited in Qiita with study ID 12274 (https://qiita.ucsd.edu/study/description/12274).

## SUPPLEMENTAL MATERIAL

Supplemental material is available online only.

**FIG S1**, PDF file, 0.2 MB.

**FIG S2**, PDF file, 0.4 MB.

**FIG S3**, PDF file, 0.5 MB.

**FIG S4**, PDF file, 0.03 MB.

**FIG S5**, PDF file, 0.03 MB.

**TABLE S1**, XLSX file, 0.01 MB.

**TABLE S2**, XLSX file, 0.05 MB.

**TABLE S3**, XLSX file, 0.01 MB.

**TABLE S4**, XLSX file, 0.01 MB.

**TABLE S5**, XLSX file, 0.01 MB.

## ACKNOWLEDGMENTS

We thank Personalbio (www.personalbio.cn) for help in sequencing analysis.

We declare that we have no competing interests.

We thank South China Agricultural University and Department of Education of Guangdong Province (2018KTSCX021) for financial support.

D.N. and G.-H.C. collected dust samples in hotels. D.N. and Y.S. designed the project. X.F., Y.L., Q.Y., and Y.S. carried out the analysis. X.F., Y.D., X.Z., D.N., and Y.S. drafted the manuscript. All authors read and approved the final manuscript.

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
