## [Reviewer comments · mSystems]

Continental-scale microbiome study reveals different environmental characteristics determining microbial richness and composition/quantity in hotel rooms

Xi Fu, Yanling Li, Qianqian Yuan, Gui-hong Cai, Yiqun Deng, Xin Zhang, Dan Norbäck, and Yu Sun

Corresponding Author(s): Yu Sun, South China Agricultural University

Review Timeline:

Submission Date:	February 6, 2020
Editorial Decision:	March 29, 2020
Revision Received:	March 30, 2020
Accepted:	May 3, 2020

Editor: Robert Beiko

Reviewer(s): The reviewers have opted to remain anonymous.

Transaction Report:

DOI: <https://doi.org/10.1128/mSystems.00119-20>

Reviewer #2

Summary (major findings, overall impression, major shortcomings)

This paper describes a large-scale survey of the ‘hotel microbiome’ which has not been done before. The authors looked at bacterial and fungal community distributions as well as fungal quantities (from a previous study) and correlated with environmental/metadata. They found that latitude, among others depending on the analysis, was a significant factor in differentiating microbial communities across hotel rooms. They also identified *Aspergillus* as a main fungal player which was abundant in low latitude hotels. They conclude that these results offer insights into assessing hotel health effects. The manuscript gives new insight into what persists in hotels and used statistics to correlate the fungal and bacterial communities with various measured factors (hotel type, etc.). Overall, it's an impressive survey across a large number of hotels but it did not address any hypotheses going into the study about hotel microbiomes. It was confusing and not clear as to what statistics were used with what data type (since there were so many different statistical analyses conducted) and what the overall takeaways were supposed to be – what's important got lost. The authors sometimes overstate the correlations with overall conclusions /claims throughout the manuscript that really need more data or experiments to fully back up. Only the area above the door frame was sampled and it's unclear how much of that represents the microbiome of the entire hotel room. I am also concerned that the negative controls were assessed by gel but were not sequenced – these should always be sequenced in microbiome studies for the built environment (may not see anything on the gel, but there could still be contaminants).

Thanks for the reviewer's professional comments! We think these comments significantly improve the quality of the manuscript and help us correct errors and mistakes in the manuscript. We address the following specific comments below.

Comments (numbered)

Overall, please be specific in what analyses and results are bacterial, fungal, or ‘microbial (both)’, and be consistent with the use of richness and community vs. alpha and beta diversity. It will make the manuscript much easier to read.

Thanks for the suggestions. We corrected the unclear words and phrases in the manuscript. We removed alpha and beta diversity in the manuscript and used richness and composition variation.

Also, shouldn't multiple comparison error corrections be used for (at least some) the statistics? Such as Bonferroni?

In this study, we conducted bivariate analysis first to screen potential associated environmental characteristics. Then, all the environmental characteristics with marginally significant associations ($p < 0.2$) were put into a multivariate model with a

stepwise approach. The environmental characteristic with the lowest p-value in the bivariate analysis was input first in the multivariate model, and the environmental characteristic with the second lowest p-value was input next, and so on (inclusion level $p = 0.2$). If the newly added characteristic did not improve the model ($p > 0.1$), the characteristic was removed from the multivariate model and the next characteristic was tested. Thus, the final model is a single multivariate model, which did not have the multiple correction problem. The reason why we did not use a multivariate model from the beginning is that there were 16 environmental characteristics, and putting so many characteristics together could lead to an overfitting problem. Even though there could be some potential false positives in the bivariate, these associations can be adjusted in the final multivariate model. The stepwise approach will put the dominant characteristics first in the model and use the dominant characteristics to adjust the newly added variables.

Overall, we rewrote the whole section. We removed the section describing the results of bivariate associations and Table 1 in the manuscript. The data for bivariate analyses were all put in the supplementary tables. Now only the final multivariate model is described in detail.

Maybe I missed it, but was Figure S7 ever referenced in the manuscript?

Sorry, we forgot to refer to Figure S7 in the manuscript. We corrected the labeling of the supplementary figures.

Line 40, Are these normalized by sequencing depth? Also, operational taxonomic units can be defined here as the abbreviated OTUs.

The bacterial and fungal sequencing data were rarefied in the same depths of 27,000 reads. We added “OTUs” in the sentence. The sentence revised as “Similar numbers of bacterial (4,344) and fungal (4,555) operational taxonomic units (OTUs) were identified in the same sequencing depth, but fungal taxa showed a local distribution compared with bacterial taxa.”

Line 143, What is a professional hygienist? I think it would be more important to state that how the samples were collected (e.g., aseptically)

A professional hygienist is an academic person specialized in environmental sampling. The samples were collected aseptically, by swabbing 60 cm² of the doorframe area. We revised the sentence in the text “Dust swab samples were collected by a professional hygienist, an academic person specialized in environmental sampling.” “We collected dust samples in one room in each hotel by using dry cotton swabs to swab the upper half of the doorframe. The swab was designed for medical DNA sampling, and each swab was packaged in a DNA free and sterile plastic vessel (Copan Innovation, Brescia, Italy; <http://www.copanitalia.com>). Two samples were collected in each

room with a swabbing area of 30 cm² (1 x 30 cm) on the left- and right-hand sides of the doorframe. Each swab was rotated slowly and moved back and forth 3 times over the surface. The swabs were stored in a -80°C freezer after sampling. One swab was used for amplicon sequencing in this study, and one swab was used for quantitative PCR in a previous study [36].”

Line 146, Why was only the door frame sampled? How representative is this of the hotel room?

The main reason is that other sites, such as floor or furniture surfaces, are frequently cleaned by hotel staff. Thus, the dust samples on the floor may only represent microbiome composition in the short-term and also affected by cleaning methods. We want to collect dust samples that represent relatively long-term exposure to hotel guests and thus decide to sample on the top of the door frame, which is not cleaned in the standard cleaning routine of hotel rooms. Other sampling sites, such as the top of the mirror cabinet could also be possible. But these sites are not standardized in hotels; some hotels install these cabinets whereas the others are not. Also, the surface material types of the mirror cabinet could be different in different hotels, which can also change the microbiome composition. Overall, we think the top of the door frame is a good choice for sampling microbial exposure in the settled air dust. Future studies, with other passive sampling strategies or active air sampling strategies, such as by an air vacuum pump, could bring a more comprehensive view of indoor microbial exposure for hotel guests.

Line 150, Was the type of material noted (paint type?)

Only the floor surface type of material is noted and presented as one of the environmental characteristics in the study.

Line 151, Was visible mold assessed as presence/absence? Or is this based on some rubric/scale?

It is yes/no, no scale. We added this information in Materials and Methods.

Line 151, I think it should be stated, “Pearson’s correlation was conducted to detect and reduce collinearity”. Also, could it be stated here how many were removed due to collinearity (if its in a supplementary table, please state)

We changed the sentence according to the reviewer’s suggestion. We also expanded the paragraph to describe the criteria for removing certain environmental characteristics in details

“Twenty-nine environmental characteristics were assessed and recorded at each sampling site (Table S1). Eight environmental characteristics were only detected in very few hotels and thus removed from further analysis. Pearson’s correlation was

conducted to detect and reduce collinearity ($\rho > 0.7$), and four environmental characteristics were removed. A final set of seventeen environmental characteristics were kept for further analysis, including latitude, surrounding traffic, quality of the interior (textiles, walls and furniture), and visible mold (Table 1).”

Line 155, The supplementary figures are off – they do not line up with the S1, S2 numbering in the manuscript. I do not see the gel image. Overall, this is concerning that the negative controls were not sequenced. Looking at a gel image is not always accurate, and it would be more appropriate to sequence. I would highly recommend for this to be done.

Sorry that supplementary figures 1 and 2 were not included in the previous version. Now it is fixed. The negative controls were not sequenced. We should have done this as a quality control step in the sequencing process, but now we cannot do it as the sequencing company has already finished the project and the gel was not kept afterward. Although this is an issue of the study, we would also like to argue that sequencing the negative controls in environmental dust samples may not as critical as in gastrointestinal studies. First, the dust samples are dry and contain a very limited amount of nutrition compared with fecal samples, thus the mass proliferation of contaminating species is less likely compared with fecal samples. Second, for gut microbiota studies, identifying environmental microbes in the gut microbiome data give a good hint of contaminations in samples. The solution can be simply removing the contaminating species detected in the negative control. However, the common laboratory contaminating microbes are also common microbes in the indoor environment. Thus, for indoor microbiome studies, even these microbes are sequenced and detected in negative controls, it is challenging how to deal with them, and it is not appropriate to just remove them in the sequencing data. Third, a recent study reported that well-to-well contamination is common during microbiome sequencing. The microbes detected in negative controls may not from laboratory contaminations, but rather from sequenced samples in the adjacent wells (Minich 2019, Microbiome). Thus, it suggests that negative control could also be misleading, and caution should be taken in interpreting the results. In this study, the gel image indicates that there is no high amount of DNA detected in the negative control. Even the approach is less sensitive compared to sequencing, at least we know that there is no mass contamination in the laboratory.

Lines 161-165, Are the OTUs found in just 1-2 samples also the most abundant? Same for the bacterial OTUs, were the 43 OTUs found in all samples also fairly abundant?

The widespread OTUs were mainly the high abundant OTUs. We modified the sentence as “Bacterial and fungal samples had distinct OTU distribution patterns (Figure 1A, 1B). Most fungal OTUs were present in a few samples; approximately half of the OTUs were found in 1 or 2 samples. The widely spread fungal OTUs were

mainly the high abundant fungal OTUs. For example, *Candida albicans* was found in all samples, and the mean abundance of the species was 4.1%. *Aspergillus* was found in 66 samples, and the mean abundance of the genus was 26.5%. The bacterial OTUs were more widespread on a regional or continental scale; more than half of the OTUs were found in 10 or more samples and 43 OTUs were found in all samples. The most widely spread bacterial OTUs were also mainly the high abundant bacterial OTUs, such as *Pelomonas* and *Ralstonia*. The differences between bacterial and fungal distribution were that the number of widely distributed bacterial species is much higher than fungal species, and a large proportion of fungal species had restricted distribution in 1 or 2 samples.”

Lines 169-170, Are these values and standard deviations across all hotel rooms sampled?

Yes, these are mean and standard deviation across all hotel rooms. We also added this information in the sentence as “For bacteria, the dominant phylum was Proteobacteria (mean \pm standard deviation across all samples, 71.8% \pm 11.2%)”.

Lines 175-177, Could more information be provided in regards to the microbiome novelty score? It is just that the curated database is biased away from built environment studies?

Line 178-179, So is this statement for bacteria only or also for the fungi?

We re-wrote this paragraph to describe the search engine. The whole paragraph is about bacteria. “We searched bacterial compositional similarity of our hotel samples against a newly constructed and curated public database Microbiome Search Engine, which contains the compositional information of more than 230,000 bacterial microbiome samples across a variety of environment, including human and animal gastrointestinal tract, nose and skin, soil, river, food, water, sand and also building environment [44]. The engine calculates a score for a query sample, and a high novelty score (> 0.12) represents a low similarity to previous samples in the database. The similarity scores were > 0.12 for 70% of the hotel samples, indicating high compositional novelty in our hotel bacterial dataset (Table S4). The most similar bacterial microbiome samples were mainly from building and human nasopharynx and skin environments.”

Line 186, Could the low, mid, and high latitude be indicated on the figure?

The label of low, mid and high latitudes was added in Figure 1.

Line 192, In another part of the manuscript (methods, line 518), it makes it seem like a Bray-Curtis was used as the dissimilarity distance for bacteria and fungi. Then in Table S6, it states that weighted UniFrac distance was used. Please clarify throughout.

The Adonis analyses were based on a weighted UniFrac distance matrix for bacteria and a Bray-Curtis dissimilarity matrix for fungi. We changed the sentence in the Methods section. UniFrac distance was suggested to a better matrix, which integrated phylogenetic information in the calculation. However, the ITS sequences for fungi did not produce a solid phylogenetic distance as 16S for bacteria, and thus, the Bray-Curtis matrix was used for fungi.

Line 195-197, The percent explained variability is also not very high for axis 1 so I'm not sure they can conclude it's a single dominant factor? The same could be said about the bacterial PCoA based on the explained variability of axis 1 vs. axis 2.

The PCoA analysis was replaced by RDA analysis. In the RDA analysis, the latitude was a strong factor for fungal composition variation (axis 1 eigenvalue 55.1%).

“The effects of environmental characteristics on the microbial community variation were further illustrated by the Redundancy Analysis (RDA; Figure 3). The environmental characteristics identified to be associated with bacterial and fungal community variation ($p < 0.1$, Adonis) were projected on the community variation plot. Latitude was a variable explaining bacterial variation along RDA axis 1 and 2. Floor type explained bacterial variation along RDA axis 1. For the fungal community, latitude was an important characteristic explaining variation along RDA axis 1, which accounted for 55.1% of eigenvalues. Other environmental factors, such as floor type and quality of the interior, mainly explained fungal variation along RDA axis 2.”

Line 202-203, Is this rarefaction (27,000 reads) for bacteria and fungi?

Yes. We added it in the sentence. “The bacterial and fungal sequencing data were both rarefied to the depth of 27,000 reads, and all following analyses were conducted at this depth.”

The reference to Table 1 in this sentence is confusing, because its only showing alpha diversity (OTUs)— can this be clarified? Also, Table 1, this table is really hard to interpret, seems like some of the lines are off at the top. Could just the significant p values be indicated, and then the others be put in the supplement?

Line 207, species richness, based on OTUs? Could the name of the test be indicated here too for clarity?

Line 209, I don't know if they can state, “latitude is an important factor shaping the bacterial and fungal community variation.” The R2 values are very small and there could be other factors at play. This sentence should be revised.

Lines 210-213, Where is this analysis? What is meant by ‘were also associated’? Is there some statistical test used to back this up?

These comments can be discussed together. We removed the majority of the

paragraphs describing bivariate analysis between environmental characteristics and microbial data and moved the Table 1 to supplementary. The main purpose of bivariate analyses was to screen potential associations ($p < 0.2$) and put them step-wisely into the multivariate model. Thus, multivariate analysis is the important analysis in this study, and it is redundant to describe all the bivariate analyses in detail. We also discussed statistic choice here in a few sentences. We used simple KW tests in the bacterial/fungal richness and fungal quantity analysis. However, for the community variation analysis, Adonis was used as compositional data are multi-dimensional and cannot be analyzed by the KW test. Also, the bacterial and fungal analysis were all based on observed OTUs at the depth of 27,000 reads. The modified paragraphs were pasted below:

“Association analyses between environmental characteristics and hotel microbiome

As a next step, associations were further quantitatively analyzed between environmental characteristics and microbiome datasets, including bacterial and fungal richness and composition variation (Table 1). The data of absolute quantification of fungal DNA were also included in the analyses, which was assessed by quantitative PCR by using two sets of primers in our previous studies in these hotel rooms [34]. The two primers targeted different fungal amplicon regions (ITS1 and 28S rRNA gene) and captured a wide range of indoor fungal species. We first conducted the bivariate analyses between each environmental characteristics and microbial data (Table S7-S9). The associations between environmental characteristics and bacterial/fungal richness (based on the number of observed OTUs) and fungal DNA were conducted by the Kruskal-Wallis test. The multidimensional compositional variation data cannot be analyzed by simple statistics such as Kruskal-Wallis test, and thus were analyzed by Adonis. The environmental characteristic with p -value < 0.2 in the bivariate analysis were included in the final linear multivariate analysis model by the forward stepwise approach (for bacterial/fungal richness and fungal DNA) and multivariate Adonis analysis (for bacterial/fungal compositional variation; Table 1).”

Line 215, I understand that this references the previous study in terms of the fungal DNA quantity measurements, but since the data are used in this study, it needs to be better described, especially in the methods (at least briefly) without having the reader going back and having to read through the other manuscript.

We added a new paragraph describing the targets of the two sets of primers. “Our previous study used two sets of primers to quantify absolute fungal DNA in these hotel rooms [34]. The first primer set targeted the fungal ITS1 region and captured a wide range of indoor fungi (>530 species), including 7 *Acremonium*, 61 *Alternaria*, 86 *Aspergillus*, 38 *Cladosporium*, 14 *Curvularia*, 27 *Eupenicillium*, 8 *Fusarium*, 17 *Neosartorya*, 15 *Paecilomyces*, 157 *Penicillium*, 9 *Rhinochadiella* and several other species. The complete list of targeted species can be referred to [46]. The second

primer set targeted fungal 28S rRNA and captured mainly 37 *Aspergillus*, 62 *Penicillium*, 14 *Eupenicillium* and several other species (in total >140 species). We named the two fungal quantifications fungal DNA 1 and fungal DNA 2 (Table S7 and Table 2) in this study.

”

Line 216, Does the first primer set also capture *Aspergillus* and *Penicillium* taxa? Are both primer sets from reference 46?

The first primer set can also capture *Aspergillus* and *Penicillium*. However, as the two sets of primers capture two different amplicon regions (ITS and 28S rRNA gene), the *Aspergillus* and *Penicillium* species captured by the two sets of primers could be different. Indeed, from the NCBI sequence database, we estimated that at least 86 *Aspergillus* and 157 *Penicillium* species should be captured by primer 1, and 37 *Aspergillus* and 62 *Penicillium* species should be captured by primer 2. Thus, the two primer sets represent two ways to quantify fungal DNA quantity. Both primers were from reference 46.

Line 219, Table S7, Why is there a ratio of DNA2/DNA1? Was this used for the statistics? This is not well described in the text.

This is not very useful and also a bit misleading. We removed this in Table S7.

Line 226, Is this two separate analyses – both forward step-wise linear multivariate regression and Adonis analysis that were done?

Yes, two separate analyses were conducted, including linear multivariate regression and Adonis. Regression is one of the golden standards in association studies, but for the same season discussed previously (microbial community variation data was multidimensional matrix data), Adonis was used in the community variation analysis.

Line 229-230, These R² values seem really small? Is this the Adonis analysis?

Yes, the R² values were from Adonis. Probably some other environmental characteristics not recorded in this study also involved in shaping microbial community variations. Also, previous microbiome studies showed that random variation could affect the microbiome dataset. These randomnesses can also account for a large proportion of variations in the data.

Line 232, Confused as to where these values of beta and CI are coming from? Seems like this is defined later (lines 236-237) but should be defined earlier.

Beta and CI were from linear regression. Now we define it before the analysis.

Line 240, I don't think they can state that all these factors significantly increased fungal DNA quantity, all that's known is there is a statistically relevant association.

We removed all words "increased" and "decreased" in the paragraph. Instead, now we wrote as "was positively/negatively associated with" in these sentences.

Lines 235-240 and 243-245, Why are these factors broken out into different separate analyses? This section was incredibly unclear and overall it's difficult as the reader to keep track of all of the different associations and factors between the bivariate and multivariate analyses and between alpha and beta diversity, quantity and taxa.

Regression is one of the golden standards in association studies, but for the same season discussed previously (microbial community variation data was multidimensional matrix data), Adonis was used in the community variation analysis. We rewrote the majority of these paragraphs to clarify the rationale for different multivariate analyses.

"The environmental characteristic with p -value < 0.2 in the bivariate analysis were included in the final linear multivariate analysis model by the forward stepwise approach (for bacterial/fungal richness and fungal DNA) and multivariate Adonis analysis (for bacterial/fungal compositional variation; Table 1).

For bacteria, latitude was the strongest factor associated with compositional variation ($p < 0.001$, $R^2 = 0.10$, Adonis), followed by floor type and quality of the interior ($p < 0.05$, $R^2 = 0.06$ and 0.04 , respectively). Recent redecoration and the presence of mechanical ventilation were negatively associated with bacterial richness in the hotel rooms (linear regression coefficient $\beta = -111.77$, 95% confidence interval, CI: -170.00 to -44.55 ; $\beta = -70.92$, 95% CI: -136.71 to -5.12). Thus, bacterial richness and compositional variation were associated with different environmental factors.

For fungal, latitude was also the strongest predictor of compositional variation ($p < 0.001$, $R^2 = 0.14$, Adonis), and was significantly associated with fungal quantity ($\beta = -0.40$, 95% CI: -0.62 to -0.18 , linear regression). Similarly, proximity to the sea, low quality of the interior and visible mold were significantly or marginally significantly associated with fungal community variation ($p = 0.03$, 0.07 and 0.07 and $R^2 = 0.03$, 0.02 and 0.02 , respectively). All these factors were positively associated with fungal DNA quantity ($\beta = 0.82$, 95% CI: 0.37 to 1.27 ; $\beta = 0.39$, 95% CI: 0.19 to 0.59 ; and $\beta = 0.47$, 95% CI: 0.09 , 0.85 , respectively) but did not affect fungal richness. Two factors were significantly associated with fungal richness. The floor type of wall-to-wall carpet was negatively associated with fungal richness ($\beta = -25.71$, 95% CI: -47.60 to -3.82). Proximity to roads with heavy traffic was negatively associated with fungal richness ($\beta = -63.65$, 95% CI: -127.05 to -0.25). Thus, the factors associated with fungal community variation were also associated with fungal absolute quantity, while fungal richness was affected by other environmental characteristics."

Line 246, Microbial alpha diversity is stated, but isn't this fungal alpha diversity?

We changed it to "fungal richness".

Line 262, How was dampness assessed? In the room or on the sampled door frame?

Dampness was assessed as visible dampness anywhere in the room. No dampness was observed on the sampled doorframes. We added this in the Materials and Methods section.

Line 262-263, Figure S8 shows fungal, but the sentence is referencing the bacterial genus Saccharopolyspora? There is no Figure S9.

We fixed the labeling.

Line 270-275, This just summarizes the results (and very poorly for the bacteria analyses) and does not really give a sense of the 'So what' factor of this manuscript. I feel like this could be deleted.

We deleted this paragraph according to the reviewer's suggestion.

Line 289-292, The qPCR quantification did not seem to be a significant part of this study, so I'm not sure I would emphasize that this is a major differentiator between this study and others.

We removed the sentence in the Discussion. "However, these studies did not conduct qPCR quantification; thus, the associations between environmental factors and microbial quantity were not clear." We also added a new sentence "The major finding of our study is that this is the first microbiome survey in the hotel environment", which is a major differentiator between this study and other studies.

Line 297-299, Since relative humidity was not measured in this study, that seems a bit of stretch to compare these results to the humidity chamber models from the Dannemiller lab.

Line 300, What model? What exactly is the comprehensive model put forth by this study? A cartoon model (or something like it) could be a good ending figure to put in this manuscript, to help the reader digest all of the statistical information and associated factors.

We removed the sentences from line 297-300. First, these sentences are not clear. Also, these sentences should not be here. This paragraph is mainly discussing what has been done in previous studies and what is novel in this study. But it does not aim

to discuss the effect of different environmental factors, which, including latitude, was discussed in detail in the following section “Factors structuring microbial composition variation”. Thus, these sentences were redundant and unclear and thus were removed from here.

Line 310-314, I personally don't think this is acceptable. Bottom line is that the negative control samples should have been sequenced.

Same as above. We pasted the previous response here.

Sorry we did not sequence the microbial composition of negative controls, and it cannot be done as the gels were not kept in the sequencing company. The negative controls were not sequenced. We should have done this as a quality control step in the sequencing process, but now we cannot do it as the sequencing company has already finished the project and the gel was not kept afterward. Although this is an issue of the study, we would also like to argue that sequencing the negative controls in environmental dust samples may not be as critical as in gastrointestinal studies. First, the dust samples are dry and contain a very limited amount of nutrition compared with fecal samples, thus the mass proliferation of contaminating species is less likely compared with fecal samples. Second, for gut microbiota studies, identifying environmental microbes in the gut microbiome data give a good hint of contaminations in samples. The solution can be simply removing the contaminating species detected in the negative control. However, the common laboratory contaminating microbes are also common microbes in the indoor environment. Thus, for indoor microbiome studies, even these microbes are sequenced and detected in negative controls, it is challenging how to deal with them, and it is not appropriate to just remove them in the sequencing data. Third, a recent study reported that well-to-well contamination is common during microbiome sequencing. The microbes detected in negative controls may not be from laboratory contaminations, but rather from sequenced samples in the adjacent wells (Minich 2019, Microbiome). Thus, it suggests that negative control could also be misleading, and caution should be taken in interpreting the results. In this study, the gel image indicates that there is no high amount of DNA detected in the negative control. Even the approach is less sensitive compared to sequencing, at least we know that there is no mass contamination in the laboratory.

Line 323-324, If the negative control samples weren't sequenced, how do they know that?

Sorry for the unclear sentence. We rewrote the sentences and hope it is clear. “In hotel rooms, *Aspergillus* was the dominant fungus, accounting for on average one-fourth of the total fungal load. However, the species was not evenly distributed among all samples; in 13 hotels, the abundance of *Aspergillus* was > 50%, and in approximately half of the hotels the abundance was < 10%. We also checked other sequencing

projects in the same MiSeq run and did not find a high abundance of *Aspergillus*, indicating it is unlikely to be derived from common laboratory contamination.”

Line 334-335, How do they know these were correlated? Were these measured? If so, please describe.

We did not measure RH in this study, so we do not have data for the correlation between RH and latitude/mold/proximity to the sea. I changed the sentence as “Besides low latitude, proximity to the sea and visible mold were also associated with a high abundance of *Aspergillus*. These environmental characteristics are suggested to relate to high indoor air moisture and relative humidity (RH) [10].”

Line 352-353, Are all species/strains potential allergens?

We do not know if all species produce potential allergens. The allergic effects of *Aspergillus* were mainly studied in *Aspergillus fumigatus*, but also with publications in *Aspergillus flavus* and *Aspergillus niger*. For many other *Aspergillus* species, the allergic effects are still not clear. We added the following sentences in the paragraph.

“Allergens of *Aspergillus fumigatus* have been extensively characterized by IUIS Allergen Nomenclature Committee. These allergens, including Asp f 36, Asp f 37, Asp f 1 and many others, can cause Type I and Type III hypersensitivity reactions in humans [57]. The allergic effects of other *Aspergillus* species, such as *Aspergillus flavus* and *Aspergillus niger*, were also reported [58].”

Line 362-363, Rephrase “We found that latitude did not affect the microbial richness in hotel rooms” to something like microbial richness was not correlated or was independent of latitude.

End of line 363, Make this the start of a new paragraph, A relationship between mechanical ventilation...

We changed according to the reviewer’s suggestions.

Line 371-376, I would be cautious here, as the carpet in this study was not actually sampled for fungi community analysis

Yes. We added the sentence at the end of this paragraph “As the dust was sampled from door frame rather than from carpet, the results and interpretation should be treated with caution.”

Line 384-385, I’m not sure I understand this sentence? It doesn’t make sense to me. Do they mean indoor fungi are mainly from outdoor fungi which is more geographically patterned than indoor bacteria?

We rewrote the whole paragraph and hope it is clear now. “In this study, latitude was an important factor determining both bacterial and fungal composition as shown in the Adonis analysis. The importance of latitude and geographic distance has been well documented in several microbiome studies [46, 47, 65]. For example, a national survey of 1,200 households revealed that both the bacterial and fungal composition was significantly affected by geographic range and latitude [20]. However, there are also differences between the dispersal patterns between bacterial and fungal communities. It has been reported that the fungal community showed stronger geographic patterns and dispersal limitation at a short distance [46]. We confirmed this finding by showing that most of the fungal OTUs had restricted distribution and presented in only one or two hotel rooms. It has been reported that the source of indoor fungi is mainly derived from the outdoor environment, whereas the source of indoor bacteria is more complex and affected by both the outdoor environment and indoor occupants, pets and plants [67]. Thus, the bacterial distribution in the indoor environment can be facilitated by human travel and movement compared with the fungal distribution.”

Lines 398-403, Somewhat contradictory here in stating that latitude, and not indoor factors, was important in shaping microbial community, but then in the next sentence stating indoor factors were significantly associated with bacterial composition variation. Also, referring to taxa or the beta diversity? This also just seems like a recapitulation of the results and I’m not sure what it adds. I would recommend re-working this section.

We wrote much of this paragraph to work on the logic. “The importance of global and outdoor environmental characteristics in shaping the indoor fungal composition is well supported, but the importance of indoor characteristics is still under debate. Some studies suggest that indoor characteristics are not important. A survey of indoor environments revealed that global factors, rather than building design and materials, determine the indoor fungal composition, and the indoor fungal assemblage represents a subsample of the outdoor fungal community [47]. Other studies support this by showing that most fungi may not grow or proliferate in the indoor environment; thus, the indoor environment mainly serves as a passive collector for the outdoor fungal biome [46, 47]. However, there are studies suggest that indoor characteristics are also important in shaping indoor fungal composition. A recent survey of university residences in California found that fungal composition clustered by indoor surface type, suggesting that some fungal species do grow or adhere to certain surface types [67]. In this study, we found that global characteristics, such as latitude, and indoor characteristics, such as the quality of the interior and floor surface type, were all involved in shaping fungal composition in the indoor environment.”

Line 411, Do they mean co-correlated?

Yes, we found a medium correlation between urban/rural location and proximity to a

sea. We also rewrote the sentence to make it more understandable. “In this study, we did not detect an association between urban/rural locations and microbial variations. The association was significant in the bivariate analysis but not after adjusting for latitude and proximity to the sea in the multivariate model. This could be due to the medium collinearity between proximity to a sea and urban/rural locations ($\rho=-0.54$, $p < 0.001$, Pearson’s correlation).”

Line 421, What does “were assessed by professional standards” mean?

Sampling was performed in a standardized way, by a person with education on environmental sampling and indoor environment inspections. We added the sentence “Sampling was performed in a standardized way, by a person with education on environmental sampling and indoor environment inspections. Thus, the environmental characteristics were collected more consistent than the self-reported observations of residents in some home studies.”

Line 432 “and the quantity results should be interpreted with caution”. And then in the next sentence, “variation in the sampling process should be minimal”. Can this discrepancy be clarified?

We rewrote it as “Since the hand pressure for swabbing may vary across the sampling sites, biases can be introduced in the sampling process. But we argue that the bias should be relatively small as the dust swabs were all collected by a single hygienist, and thus the sampling practice should be relatively consistent.”

Lines 434-437, Yes this is true – but without sequencing them, there is no way to know by how much.

Yes, we agree with that and remove this sentence.

Line 442, I would state fungal here instead of microbial?

We changed the sentence according to the reviewer’s suggestion.

Line 445-446, Isn’t this true of the fungi too? Even more so?

Yes, that is true. We changed the sentence as “Most microbes in hotel rooms were ubiquitous species sourced from outdoor environments instead of from human sources.”

Line 458, The samples were collected 10 years ago – when was the DNA extracted and then sequenced? Is there any concern with degradation? As well as how this affects the interpretation of the results, in terms of mold contamination, considering that cleaning practices, etc. could have changed since then?

The swab samples were stored in -80 °C freezer after sampling from 2007 to 2009. The DNA was extracted and sequenced in 2018. We did not take the samples out of freezer until the sampling in 2018, and we did not observe any signs of microbial contamination in the freezer. Also, the sequencing results from the hotel samples and other samples stored in the same freezer did not show signs of contamination. Overall, we think the samples and sequencing results should reflect the situation in the hotel rooms at that time.

Line 459, Curious why cotton and not nylon swabs were used? Were these sterile?

Dry cotton swab, sterile and DNA free, delivered by the company in a DNA free sterile plastic vessel. It is designed for medical DNA sampling (Copan Innovation, Brescia, Italy; <http://www.copanitalia.com>). We added the sentence in Materials and Methods section " The swab was designed for medical DNA sampling, and each swab was packaged in a DNA free and sterile plastic vessel (Copan Innovation, Brescia, Italy; <http://www.copanitalia.com>).".

Line 461-462, How long between sampling before being transferred to -80C? Could community be affected on the swab in transit?

The swabs were dry and were kept at room temperature until returning to Uppsala University, Sweden where the samples were stored at -80C. From a few days up to a week kept at room temperature. The swab was designed for medical DNA sampling and was packaged in a plastic vessel after sampling.

Line 481, typo, quantity

Corrected.

Line 485, primer reference?

We added the primer references in the text. "The universal forward primer 338F (ACTCCTACGGGAGGCAGCA) and reverse primer 806R (GGACTACHVGGGTWTCTAAT) [69] were used for bacterial 16S rRNA gene V3-V4 region amplification, and the amplification region was 480 bp in length. The forward primer ITS5 (GGAAGTAAAAGTCGTAACAAGG) and reverse primer ITS2 (GCTGCGTTCTTCATCGATGC) [70] were used for fungal ITS1 region amplification, and the amplification region length was 250 bp."

Line 511, What method was used for picking OTUs? Why weren't ASV's used (DADA2)?

We used `pick_open_reference_otus.py` in QIIME (v1.8.0) for OTU picking.

In the methods, I would briefly describe the fungal quantities and how they were collected - other manuscript can be referenced, but there needs to be some mention of it here.

We added one sentence in the sample collection section. “One swab was used for amplicon sequencing in this study, and one swab was used for quantitative PCR in a previous study [34].”

We also added a new paragraph describing how the qPCR was conducted.

“Quantitative PCR of fungal DNA

Our previous study used two sets of primers to quantify absolute fungal DNA in these hotel rooms [34]. The first primer set targeted the fungal ITS1 region and captured a wide range of indoor fungi (>530 species), including 7 *Acremonium*, 61 *Alternaria*, 86 *Aspergillus*, 38 *Cladosporium*, 14 *Curvularia*, 27 *Eupenicillium*, 8 *Fusarium*, 17 *Neosartorya*, 15 *Paecilomyces*, 157 *Penicillium*, 9 *Rhinochadiella* and several other species. The complete list of targeted species can be referred to [80]. The second primer set targeted fungal 28S rRNA and captured mainly 37 *Aspergillus*, 62 *Penicillium*, 14 *Eupenicillium* and several other species (in total >140 species). We named the two fungal quantifications fungal DNA 1 and fungal DNA 2 (Table S7 and Table 2) in this study.”

Line 532, Are the scripts used also deposited?

No, the scripts were not deposited. The regression analyses were conducted in SPSS, an easy-to-use graphic software. Adonis analyses were conducted in the vegan package in R, which is also quite simple and straightforward analyses. Thus, we did not deposit scripts.

Reviewer #3

Thanks for the reviewer’s professional comments! We think these comments significantly improve the quality of the manuscript and help us correct errors and mistakes in the manuscript. We address the following specific comments below.

General Comments

1. This manuscript is well written and polished, save for some incorrect labeling of supplementary figures.
2. The introduction is comprehensive and well cited.
3. The conclusion includes necessary discussion of the limitations of this study, such as the lack of replicate hotel rooms.

Thanks!

4. I think more could be done with the data. In particular, there could be more done with the qPCR data. By combining qPCR data with the fungal amplicon data, the authors could transform their fungal relative abundances into absolute abundances, which would allow them to make stronger claims about the actual increase or decrease in certain fungal taxa. Such as in Zhang et. al 2017 "Soil bacterial quantification approaches..."

We conducted the absolute quantification as the reviewer suggested, and found that all significantly differently identified by the relative abundance were also detected by the absolute abundance approach. We added a new paragraph describing the results.

"Previous studies reported that the absolute quantification approach is more accurate than the relative quantification approach in identifying the associations between microbes and phenotypes [44]. Thus, we conducted the same analysis between environmental characteristics and specific microbes based on the absolute quantification, which was calculated as multiplying the relative abundance of the fungi and qPCR data (Table S10). We confirmed that all significant differences detected by the relative abundance were also detected by the absolute abundance approach.

"

5. There could be more thorough analysis of some of the effects of latitude, such as RDA to look at how taxa vary over the latitude gradient.

6. I believe the first figure could be made simpler and more informative.

We changed the two figures. Please see the details below.

Introduction

I think this introduction is well written and well cited. I have only one comment on this section.

Line 100-105: Do you have citations for hotel room numbers and standard hotel cleaning policies? I think having citations and more exact numbers would strengthen your argument in this section.

We added two references in the manuscript. We got the number of hotel rooms from booking.com. This is almost certainly an underestimation of the true number, but probably the best we can have now. The cleaning policies were cited from the Standard Operating Procedure for hotels website. The revised paragraph was pasted below.

"Hotels are common public environments for guests and hotel staff. There are at least 22 million hotel rooms in the world [31], and billions of guests and travelers stay in

hotels each year. Thus, there is a public health concern regarding hotels' hygiene standards and practice. Unlike household residences, each hotel room is shared by many guests, but many influencing factors are also controlled. For example, many hotels use standard cleaning procedures and ventilation systems for air exchange [32], and no pets are allowed in hotel rooms so a major source of the indoor microbiome is also controlled."

Results

Line 155: Figures S1 and S2 appear to be rarefaction curves, not gels. Further, I cannot find gel images in the entire manuscript. The authors should add this information.

Line 159: Figure S3 and S4 appear mislabeled, please fix this.

Sorry about the mislabeling of supplementary figures. We corrected them in the manuscript.

Line 204 and 207: You should add some sort of measure of the effect size, as it is difficult to understand the p-values without some idea of how large the difference in diversity was.

We rewrote the whole section as it is not very clearly presented. We removed the section describing the results of bivariate associations and Table 1 in the manuscript, as the bivariate analysis was only used to screen potential associations to put in the final multivariate model. The data for bivariate analyses were all put in the supplementary tables. Now only the final multivariate model is described in detail, and the effect size was included. The new section was pasted below.

"Associations between environmental characteristics and hotel microbiome

As a next step, associations were quantitatively analyzed between environmental characteristics and microbiome datasets, including bacterial and fungal richness and compositional variation (Table 1). The data of absolute quantification of fungal DNA were also included in the analyses, which was assessed by quantitative PCR by using two sets of primers in our previous studies in these hotel rooms [36]. The two primers targeted different fungal amplicon regions (ITS1 and 28S rRNA gene) and captured a wide range of indoor fungal species. We first conducted the bivariate analyses between each environmental characteristics and microbial data to screen for potential associated environmental characteristics (Table S7-S9). The associations between environmental characteristics and bacterial/fungal richness (based on the number of observed OTUs) and fungal DNA were conducted by the Kruskal-Wallis test. The multidimensional microbial composition data cannot be analyzed by simple statistics such as the Kruskal-Wallis test, and thus were analyzed by Adonis. The environmental characteristic with p-value < 0.2 in the bivariate analysis were included in the final linear multivariate analysis model by the forward stepwise approach (for

bacterial/fungal richness and fungal DNA) and multivariate Adonis analysis (for bacterial/fungal compositional variation; Table 1).

For bacteria, latitude was the strongest factor associated with compositional variation ($p < 0.001$, $R^2 = 0.10$, Adonis), followed by floor type and quality of the interior ($p < 0.05$, $R^2 = 0.06$ and 0.04 , respectively). Recent redecoration and the presence of mechanical ventilation were negatively associated with bacterial richness in the hotel rooms (linear regression coefficient $\beta = -111.77$, 95% confident interval, CI: -170.00 to -44.55; $\beta = -70.92$, 95% CI: -136.71 to -5.12). Thus, bacterial richness and compositional variation were associated with different environmental factors.

For fungal, latitude was also the strongest predictor of compositional variation ($p < 0.001$, $R^2 = 0.14$, Adonis), and was significantly associated with fungal quantity ($\beta = -0.40$, 95% CI: -0.62 to -0.18, linear regression). Similarly, proximity to the sea, low quality of the interior and visible mold were significantly or marginally significantly associated with fungal community variation ($p = 0.03$, 0.07 and 0.07 and $R^2 = 0.03$, 0.02 and 0.02 , respectively). All these factors were positively associated with fungal DNA quantity ($\beta = 0.82$, 95% CI: 0.37 to 1.27; $\beta = 0.39$, 95% CI: 0.19 to 0.59; and $\beta = 0.47$, 95% CI: 0.09, 0.85, respectively) but did not affect fungal richness. Two factors were significantly associated with fungal richness. The floor type of wall-to-wall carpet was negatively associated with fungal richness ($\beta = -25.71$, 95% CI: -47.60 to -3.82). Proximity to roads with heavy traffic was negatively associated with fungal richness ($\beta = -63.65$, 95% CI: -127.05 to -0.25). Thus, the factors associated with fungal community variation were also associated with fungal absolute quantity, while fungal richness was affected by other environmental characteristics.

Line 209-210: Why was latitude grouped this way? Couldn't one plot the raw latitude value and calculate an effect using Redundancy Analysis (RDA) to look at the gradient, and its effects on bacterial abundances? Explanation and examples here: <http://mb3is.megx.net/gustame/constrained-analyses/rda>

We made a new RDA plot to replace the PCoA plot. Latitude, together with a few other environmental characteristics, were projected on the community variation plot. From the RDA plot, we can see that latitude is an important characteristic affecting fungal community variation. A new paragraph was added to describe the results.

“The effects of environmental characteristics on the microbial community variation were further illustrated by the Redundancy Analysis (RDA; Figure 3). The environmental characteristics identified to be associated with bacterial and fungal community variation ($p < 0.1$, Adonis) were projected on the community variation plot. Latitude was an variable explaining bacterial variation along RDA axis 1 and 2. Floor type explained bacterial variation along RDA axis 1. For the fungal community, latitude was an important characteristic explaining variation along RDA axis 1, which

accounted for 55.1% of eigenvalues. Other environmental factors, such as floor type and quality of the interior, mainly explained fungal variation along RDA axis 2.

”

Line 218-222: I don't see any mention of multiple hypothesis correction of p-values. This is necessary due to the large number of statistical tests you are performing on these data. For the 16 tests you are doing per amplicon, the odds are not good based on an alpha of 0.05, as there is only a $(1-0.05)^{16}$ or about 45% chance for each set that none are a false positive.

In this study, we conducted bivariate analysis first to screen potential associated environmental characteristics. Then, all the environmental characteristics with marginally significant associations ($p < 0.2$) were put into a multivariate model with a stepwise approach. The environmental characteristic with the lowest p-value in the bivariate analysis was input first in the multivariate model, and the environmental characteristic with the second-lowest p-value was input next, and so on (inclusion level $p = 0.2$). If the newly added characteristic did not improve the model ($p > 0.1$), the characteristic was removed from the multivariate model and the next characteristic was tested. Thus, the final model is a single multivariate model, which did not have the multiple correction problem. The reason why we did not use the multivariate model from the beginning is that there were 16 environmental characteristics, and putting so many characteristics together could lead to an overfitting problem. Even though there could be some potential false positives in the bivariate, these associations can be adjusted in the final multivariate model. The stepwise approach will put the dominant characteristics first in the model and use the dominant characteristics to adjust the newly added variables.

Overall, we rewrote the whole section. We removed the section describing the results of bivariate associations and Table 1 in the manuscript. The data for bivariate analyses were all put in the supplementary tables. Now only the final multivariate model is described in detail. We also added a new paragraph describing the rationale for the association analysis.

“As a next step, associations were quantitatively analyzed between environmental characteristics and microbiome datasets, including bacterial and fungal richness and composition variation (Table 1). The data of absolute quantification of fungal DNA were also included in the analyses, which was assessed by quantitative PCR by using two sets of primers in our previous studies in these hotel rooms [36]. The two primers targeted different fungal amplicon regions (ITS1 and 28S rRNA gene) and captured a wide range of indoor fungal species. We first conducted the bivariate analyses between each environmental characteristics and microbial data to screen for potential associations (Table S7-S9). The associations between environmental characteristics and bacterial/fungal richness (based on the number of observed OTUs) and fungal DNA were conducted by the Kruskal-Wallis test. The multidimensional compositional

variation data cannot be analyzed by simple statistics such as the Kruskal-Wallis test, and thus were analyzed by Adonis. The environmental characteristic with p-value < 0.2 in the bivariate analysis were included in the final linear multivariate analysis model by the forward stepwise approach (for bacterial/fungal richness and fungal DNA) and multivariate Adonis analysis (for bacterial/fungal compositional variation; Table 1).

”

Line 227: Table 3 does not exist in the manuscript, as far as I can tell. Did the authors mean to refer to Table 2?

Yes, now it is labeled as Table 1.

Line 250-258: I see no mention of multiple hypothesis correction in this passage, and testing this many genera without correction is problematic.

We conducted the Benjamini & Hochberg correction for p values. The only change was the *Devriesia* was not significantly different among latitudes ($q = 0.06$). We changed the figure accordingly.

Line 263: There appears to be an issue with the labeling of the supplemental figures.

We fixed the labeling.

Figure 1: I don't know how useful it is to have panels C-F in this figure, as it is clear that samples are very different across latitude groupings, and by averaging them together, it obscures this difference. Replacing it with the average diversity by latitude grouping would be more useful, or even plotting the samples similarly to 1-G to replace C-F. How are the samples grouped by 1-G?

We removed C-F according to the reviewer's suggestion. Also, bacterial abundance similar to 1-G was added in the figure.

Figure 2: Related to figure 1, showing the bacterial differences that drive this PCoA grouping would be useful. This could be done by showing the average abundance by latitude, or by grouping raw sample abundances by latitude.

According to your previous suggestions, we changed the figure as RDA analysis and projected the environmental characteristics to the microbial community variation. The effect of latitude is clearer.

Discussion

No specific issues, well written!

Thanks!

Methods

Line 455: Could you give more information about the rooms themselves, such as how they were chosen, etc?

Arbitrary chosen rooms, given by the hotel when checking in (no special request about the room). We added the sentence in Materials and Methods “Hotel rooms were arbitrarily chosen when checking in with no special request about the room.”

Line 455: Is it possible to map the various cities that these hotels are in? I think it would give a better idea of the scale of this sampling, and would enlighten the reader. I see that you are using R for plotting, and the package ggmap along with the latitudes/longitudes of the cities sampled could create such a map.

Thanks for the reviewer’s suggestion. We added a map of sampling locations as Figure 1 in the manuscript. We used the “rworldmap” package to visualize the map.

Line 459: Could you give more information about how sampling was conducted? For example, the what cotton swabs (type or model number), whether it was wet or dry, etc?

Dry cotton swab, sterile and DNA free, delivered by the company in a DNA free sterile plastic vessel. It is designed for medical DNA sampling (Copan Innovation, Brescia, Italy; <http://www.copanitalia.com>). We added the sentence in Materials and Methods section ” The swab was designed for medical DNA sampling, and each swab was packaged in a DNA free and sterile plastic vessel (Copan Innovation, Brescia, Italy; <http://www.copanitalia.com>).”.

Line 467-472: A numbered list here may be more comprehensible than listing all of the 16 factors, as it would be much easier to read. However, I acknowledge that may not be possible based on journal formatting.

Now we list all 28 environmental characteristics, including 12 removed characteristics, collected in this study in Supplementary Table 1.

Line 474: There should be sufficient information in this paper to understand the methods without digging through another paper.

We added more details in the sampling paragraph. “Dust swab samples were collected by a professional hygienist, an academic person specialized in environmental sampling. Samples were collected from 68 hotels in 19 European and Asian countries from October 2007 to May 2009. Hotel rooms were arbitrarily chosen when checking in with no special request about the room. We collected dust samples in one room in

each hotel by using dry cotton swabs to swab the upper half of the doorframe. The swab was designed for medical DNA sampling, and each swab was packaged in a DNA free and sterile plastic vessel (Copan Innovation, Brescia, Italy; <http://www.copanitalia.com>). Two samples were collected in each room with a swabbing area of 30 cm² (1 x 30 cm) on the left- and right-hand sides of the doorframe. Each swab was rotated slowly and moved back and forth 3 times over the surface. The swabs were stored in a -80°C freezer after sampling. One swab was used for amplicon sequencing in this study, and one swab was used for quantitative PCR in a previous study [36].”

Line 493: Were these libraries sequenced together?

Yes, a multiplex sequenced strategy was used. We revised the sentence as “Multiplex paired-end sequencing was performed according to the manufacturer’s instructions. The sequencing was conducted in the Illumina MiSeq platform and a MiSeq Reagent Kit v3 (600 cycles) at Shanghai Personal Biotechnology Co., Ltd (Shanghai, China).”

Line 502: What fraction of reads passed quality filtering?

We added the sentence in the method section. “In total, 94.3% of bacterial raw reads and 97.9% of fungal raw reads passed the quality control filtering.”

Line 505-509: It is not clear to me how the sequences were processed. Which OTU clustering method was used? USEARCH, mothur, etc?

We used UCLUST for OTU clustering. “The remaining high-quality sequences were clustered into operational taxonomic units (OTUs) with 97% sequence identity by UCLUST [76].”

Line 511-512: What version of each of these databases? What was done in the case of multiple identical matches?

We added the information in the paragraph. “A representative sequence was picked for each out by pick_open_reference_otus.py in QIIME (v1.8.0) and blasted against the Silva database (release 115) [78] for bacteria and UNITE database (release 5) [79] for fungi to obtain taxonomic classification information. For the case of multiple best hits, the sequence was annotated with the taxonomy corresponding to the lowest common ancestor.”

Line 513: What was the rarefaction depth?

We added the rarefaction depth in the text. “Rounded rarefied analysis was conducted to standardize the sequencing depth to 27,000 reads per sample for both bacterial and

fungal sequencing data.”

Line 516-519: I don't see any mention of multiple hypothesis correction of the results.

We added a sentence in the figure legend. “Error bars represent the standard error, and a Kruskal Wallis test with Benjamini & Hochberg correction was conducted to calculate p values (***) $q < 0.001$, ** $q < 0.01$, * $q < 0.05$).”

Line 519-521: Could the authors add more details about their stepwise regression approach, such as the effect of adding variables on the adjusted R-squared? Even details of the predictive power of the model itself?

Two multivariate analyses were conducted in this study, including multivariate Adonis and multiple linear regression. The stepwise approaches for the two statistics generally work in the same way. The difference is that for linear regression, SPSS software can automatically put potential explanatory variables to test whether adding the variables can significantly improve the model. Whereas for Adonis, the independent variables are added manually according to their p values. We added a few sentences in the method section to describe the procedure. “The multivariate Adonis analysis was calculated with a forward stepwise approach. The characteristic with the lowest p-value in the bivariate analysis was input first in the multivariate model, and the characteristics with the second-lowest p-value were input next, and so on (inclusion level $p = 0.2$). If the newly added characteristic did not improve the model ($p > 0.1$), the characteristic was removed from the multivariate model and the next characteristic was tested.”

We also added the sum of R^2 in the last line of Table 1 to describe the predictive power of the multivariate model. The sum of R^2 is > 0.5 for the models between environmental characteristics and absolute fungal DNA, and between 0.19 to 0.22 for models between environmental characteristics and microbial richness and community variations.

Line 533: The qiita link does not bring up a page. This may be because the authors haven't made it public yet.

It should work now. We choose the “release for public download” option. Alternatively, the data can be accessed by searching the Qiita ID.

March 29, 2020

Prof. Yu Sun
South China Agricultural University
Guangzhou
China

Re: mSystems00119-20 (Continental-scale microbiome study reveals different environmental characteristics determining microbial richness and composition/quantity in hotel rooms)

Dear Prof. Yu Sun:

Based on your response to your response to the previous referee comments, I am accepting your manuscript without further external review, conditional on your addressing the specific comments below.

General comments:

- In the future, do not use rarefaction or OTUs.
- I do not find the "not general contaminants" argument especially convincing. I appreciate that there is nothing you can do about it, but it does remain as a significant question mark.
- Do not rely too much on taxonomic names: there are many different types of *Ralstonia* for example, and the environmental isolates may be different from the ones you are finding in the hotel.

I am not asking you to make changes in response to these general comments, but they are weaknesses that you should avoid in the future.

Specific comments that must be addressed:

l.51, "Recent decoration"?

l.120, "Microbiome studies"

l. 121, "Provide"

l.124, "Quantify"

l.154, "Detected" -> "assessed"

l.155, Please quantify "very few".

l.170-171, "highly abundant"

l.177, "fungal" is the adjective, "fungi" the noun. The latter belongs here.

l.186, "We searched the bacterial community composition of our samples against ..."

l.189, "Environments"

l.190, "built environment"

l.198, "genera"

I am going to stop providing detailed feedback on grammar here. Please review the rest of the manuscript for additional corrections, e.g. proper use of plurals and other mistakes (e.g., l.229 - "confident interval").

l.220, please cite 'adonis' here.

l.230: those R2 values are really small. I know it's a subjective statement, but I would say "weakly

associated" to reflect this.

I.235-237: Ditto here. "Marginally significantly associated" is not a phrase that should be used, and I think those R2 values are so low that the associations are of little use. I recommend rephrasing to indicate that apart from latitude, other associations were minimal at best.

I.240-243: Same thing again. Those confidence intervals are extremely close to covering zero, and unless there is good alignment with a specific hypothesis I think these should either not be reported, or framed to indicate that there was not much in the way of strong relationships here.

I.297-298: "The major finding of our study is that this is the first microbiome survey in the hotel environment." This is not a finding. You should give an informative and specific summary of the key novel findings in your study.

Paragraph starting with I.315: But is the degree of contact between occupants similar between the surfaces examined in other studies and the doorframes you looked at here?

Paragraph at I.351: "Aspergillus is abundant in hotel rooms", this is only true if (i) you are basing this statement on quantitative results, and (ii) you have some baseline standard for "abundant". Unless the surface is completely sterile, there is going to be *something* on those door frames, and a high relative abundance of Aspergillus might not correlate with any sort of significant health risk.

I.377: I think it's important not to imply causation when the associations found here are so weak.

The same goes for the carpet discussion later on; that CI was very close to zero, so there's a pretty strong possibility that there is really no association with carpet.

Refs 31 - 32: Accessed when?

Table S1: Do you mean "hotspRing"?

Below you will find the comments of the reviewers.

To submit your modified manuscript, log onto the eJP submission site at <https://msystems.msubmit.net/cgi-bin/main.plex>. If you cannot remember your password, click the "Can't remember your password?" link and follow the instructions on the screen. Go to Author Tasks and click the appropriate manuscript title to begin the resubmission process. The information that you entered when you first submitted the paper will be displayed. Please update the information as necessary. Provide (1) point-by-point responses to the issues raised by the reviewers as file type "Response to Reviewers," not in your cover letter, and (2) a PDF file that indicates the changes from the original submission (by highlighting or underlining the changes) as file type "Marked Up Manuscript - For Review Only."

Due to the SARS-CoV-2 pandemic, our typical 60 day deadline for revisions will not be applied. I hope that you will be able to submit a revised manuscript soon, but want to reassure you that the journal will be flexible in terms of timing, particularly if experimental revisions are needed. When you are ready to resubmit, please know that our staff and Editors are working remotely and handling submissions without delay. If you do not wish to modify the manuscript and prefer to submit it to another journal, please notify me of your decision immediately so that the manuscript may be formally withdrawn from consideration by mSystems.

To avoid unnecessary delay in publication should your modified manuscript be accepted, it is important that all elements you upload meet the technical requirements for production. I strongly recommend that you check your digital images using the Rapid Inspector tool at <http://rapidinspector.cadmus.com/RapidInspector/zmw/>.

If your manuscript is accepted for publication, you will be contacted separately about payment when the proofs are issued; please follow the instructions in that e-mail. Arrangements for payment must be made before your article is published. For a complete list of **Publication Fees**, including

supplemental material costs, please visit our website.

Sincerely,

Robert Beiko

Editor, mSystems

Journals Department
Reviewer comments:

General comments:

- In the future, do not use rarefaction or OTUs.

Thanks for the comments. We start to use QIIME2 in our new project, and we start to learn how to process data in this new platform.

- I do not find the "not general contaminants" argument especially convincing. I appreciate that there is nothing you can do about it, but it does remain as a significant question mark.

"not general contaminant" is a too strong statement. We revised the sentence as "We also checked the sequencing projects in the same MiSeq run and found that most samples did not contain *Ralstonia* or *Pelomonas* species (data not shown), suggesting these taxa were not derived from laboratory contamination during the sequencing process."

- Do not rely too much on taxonomic names: there are many different types of *Ralstonia* for example, and the environmental isolates may be different from the ones you are finding in the hotel.

I am not asking you to make changes in response to these general comments, but they are weaknesses that you should avoid in the future.

Thanks for the suggestions. High-resolution approaches such as metagenomics and full-length amplicon sequencing are needed to disentangle taxa at the species level. Thus, I removed the sentence "it is common to find these taxa in hotel rooms". We will try to be caution in interpreting the results in future studies.

Specific comments that must be addressed:

1.51, "Recent decoration"?

We changed it to "recent redecoration".

1.120, "Microbiome studies"

1.121, "Provide"

1.124, "Quantify"

Done.

1.154, "Detected" -> "assessed"

1.155, Please quantify "very few".

We moved this sentence to the method section and revised it according to the suggestion. "Eight environmental characteristics, such as a sign of flood and hot spring in the hotel, were presented in less than 5 hotels and thus removed from further analysis."

1.170-171, "highly abundant"

1.177, "fungal" is the adjective, "fungi" the noun. The latter belongs here.

1.186, "We searched the bacterial community composition of our samples against ..."

1.189, "Environments"

1.190, "built environment"

1.198, "genera"

We corrected these errors.

I am going to stop providing detailed feedback on grammar here. Please review the rest of the manuscript for additional corrections, e.g. proper use of plurals and other mistakes (e.g., 1.229 - "confident interval").

Thanks for the effort! We went through the manuscript two times and used a grammar checking website to improve the text of the manuscript. We also made many other changes, such as remove the redundant sentences. The detailed revision can be checked in the manuscript with tracked changes. We hope most grammar mistake and unclear sentences has been revised.

1.220, please cite 'adonis' here.

We added the citation for Adonis.

1.230: those R2 values are really small. I know it's a subjective statement, but I would say "weakly associated" to reflect this.

We added "weakly associated" as suggested.

1.235-237: Ditto here. "Marginally significantly associated" is not a phrase that should be used, and I think those R2 values are so low that the associations are of little use. I recommend rephrasing to indicate that apart from latitude, other associations were minimal at best.

We added "weakly associated" and removed "marginally significantly associated" in the sentence.

1.240-243: Same thing again. Those confidence intervals are extremely close to covering zero, and unless there is good alignment with a specific hypothesis I think these should either not be reported, or framed to indicate that there was not much in the way of strong relationships here.

We also added "weakly associated" in the sentence to indicate that there were no strong relationships here.

1.297-298: "The major finding of our study is that this is the first microbiome survey in the hotel environment." This is not a finding. You should give an informative and specific summary of the key novel findings in your study.

We revised the sentence as “In this study, we reported the first microbiome survey in the hotel environment and found the concordant/discordant pattern for factors associated with microbial richness, compositional variation and quantity.”

Paragraph starting with 1.315: But is the degree of contact between occupants similar between the surfaces examined in other studies and the doorframes you looked at here?

Thanks for the suggestion. The degree of contact is an important factor determining the abundance of human-derived taxa. We expanded the paragraph to discuss the contact of occupants to microbiome composition. “The total contribution of human-associated taxa in hotels was approximately 10-15%. The proportion is very similar to previous dust samples collected from the upper door trim (11%) and indoor air (11%) (17, 52). But for dust collected from sampling sites with frequent human contact, such as doorknob or bed, the proportion of human-associated taxa can be much higher (16, 28).”

Paragraph at 1.351: "Aspergillus is abundant in hotel rooms", this is only true if (i) you are basing this statement on quantitative results, and (ii) you have some baseline standard for "abundant". Unless the surface is completely sterile, there is going to be *something* on those door frames, and a high relative abundance of Aspergillus might not correlate with any sort of significant health risk.

We used qPCR to quantify *Aspergillus* and *Penicillium* quantity for the same hotel rooms in a previous study, which is the fungal DNA 2 dataset used in this study. The concentration of *Aspergillus* and *Penicillium* ranged from 2×10^5 to 2×10^9 per square meter in hotel samples. Thus, there is a lot of *Aspergillus* and *Penicillium* on the hotel door frame. There are many studies reported that these species can pose adverse health effect to occupants, including asthma, allergy and infections. But as we did not collect health data for occupants in hotels, we cannot say that these species are correlated with any significant health risk in hotels. Thus, we removed the sentence “From a health perspective” at the beginning of the paragraph.

We also merged the paragraph with the previous paragraph discussing *Aspergillus*, and placed the paragraph discussing *Penicillium* after. We think the new order fit better with the logic flow.

1.377: I think it's important not to imply causation when the associations found here are so weak. The same goes for the carpet discussion later on; that CI was very close to zero, so there's a pretty strong possibility that there is really no association with carpet.

As suggested, we removed the implication for causation. We revised the sentence as “Mechanical ventilation is suggested to be associated with the variation of bacterial composition (68), but in this study, we found that it was negatively associated with bacterial richness but not associated with community variation.”

We also removed the carpet discussion as the CI was very close to zero. Instead, we added recent redecoration in this paragraph as it is a strong association between environmental characteristics

and bacterial richness identified in this study. But we did not find any previous publications to discuss this issue. “We also found an association between recent redecoration and lower bacterial richness in the hotel rooms ($p < 0.01$, 95% CI -179.00 to -44.55), which has not been reported in previous studies.”

Refs 31 - 32: Accessed when?

We added 2020 as an accessed date.

Table S1: Do you mean "hotspRing"?

Changed.

Overall, thanks for the suggestions and comments. The manuscript is much more improved with the help of editor and reviewers.

May 3, 2020

Prof. Yu Sun
South China Agricultural University
Guangzhou
China

Re: mSystems00119-20R1 (Continental-scale microbiome study reveals different environmental characteristics determining microbial richness and composition/quantity in hotel rooms)

Dear Prof. Yu Sun:

Your manuscript has been accepted, and I am forwarding it to the ASM Journals Department for publication. For your reference, ASM Journals' address is given below. Before it can be scheduled for publication, your manuscript will be checked by the mSystems senior production editor, Ellie Ghatineh, to make sure that all elements meet the technical requirements for publication. She will contact you if anything needs to be revised before copyediting and production can begin. Otherwise, you will be notified when your proofs are ready to be viewed.

Sincerely,

Robert Beiko
Editor, mSystems

Journals Department
Fig S5: Accept
Table S2: Accept
Table S1: Accept
Table S3: Accept
Fig S2: Accept
Fig S4: Accept
Fig S3: Accept
Table S4: Accept
Table S5: Accept
Fig S1: Accept